# Uncertainty Regions for Multi-Target Regression via Input-Dependent Conformal Calibration

## Abstract

We consider the problem of provable and effective uncertainty quantification (UQ) for multi-target regression tasks where we need to predict multiple related target variables. This is important in many safety-critical applications in domains including healthcare, engineering, and finance. Conformal prediction (CP) is a promising framework for calibrating predictive models for UQ with guaranteed finite sample coverage. There is relatively less work on multi-target CP compared to single-target CP, and existing methods tend to produce large prediction regions that are not useful in real-world applications. This paper proposes a novel approach referred to as *Adaptive Prediction Regions (APR)* to produce provably smaller prediction regions by exploiting heterogeneity in the input data. APR is inspired by the principle behind localized CP for single-target Guan (2023) and extends it to multi-target settings. The key idea behind APR is to perform adaptive calibration by assigning differential weights to multi-dimensional calibration examples based on their similarity to a test input. We theoretically analyze APR and show that it (a) achieves finite-sample coverage guarantees; and (b) constructs smaller prediction regions. Our experiments on diverse real-world datasets with various numbers of targets show that APR outperforms existing methods by producing significantly smaller prediction regions (achieving up to 85.51% reduction in region area) over state-of-the-art multi-target CP methods.

## 1 Introduction

Many real-world applications across domains such as healthcare, engineering, and finance involve predicting multiple related target output variables (aka multi-target regression). For example, in Patient Monitoring using wearable devices, accurately predicting both heart rate and blood pressure is essential LaFreniere et al. (2016); Moseley & Linden (2006). Similarly, in engineering, predictive maintenance systems for industrial equipment rely on models that can jointly predict vibration levels, temperature, and operational efficiency to prevent costly failures Compare et al. (2020). Advances in machine learning have enabled us to develop predictive models with high accuracy for multi-target regression tasks. However, high-stakes applications such as healthcare require more than just accurate predictions; they demand trustworthy and theoretically sound uncertainty quantification to enable safe and reliable decision-making by clinicians. For example, a prediction/uncertainty region in the multi-dimensional space that covers the true multi-target output with high probability (e.g., 95%). Conformal prediction (CP) Vovk et al. (2005); Romano et al. (2019); Guan (2019); Angelopoulos & Bates (2021); Vazquez & Facelli (2022); Angelopoulos et al. (2023) is a promising framework for achieving such provable uncertainty quantification (UQ). CP relies on a calibration approach given a black-box predictor and user-specified coverage $1-\alpha$ (e.g., 95%) to construct prediction intervals and sets that contain the true output with probability $(1-\alpha)$ for regression and classification tasks, respectively. While localized conformal prediction (LCP) Guan (2023) provides a powerful theoretical framework for input-dependent calibration in the single-target setting, our work can be viewed as a multi-target, empirically grounded extension of this idea, showing that localized calibration remains effective and tractable on a broad suite of real-world multi-output regression tasks.

Much of the existing work on CP focuses on single-target regression, and there is little work on CP for multi-target regression tasks. A naive approach for multi-target tasks is to apply CP to each target output

independently, but it can result in highly conservative (aka large) prediction/uncertainty regions, as it doesn't exploit the existing correlations between multiple target variables. Directional Quantile Regression (DQR) approach leverages the correlations among target variables to avoid their unlikely combinations in the prediction region Boček & Šiman (2017); Charlier et al. (2020). The spherically-transformed DQR approach (henceforth SOTA) Feldman et al. (2023), currently the leading conformal prediction method for multi-target regression, leverages a conditional deep generative model to learn representations of the target variables and thereby enhance DQR. However, its main limitation is that the resulting prediction regions are excessively large, making them impractical for real-world use. In the healthcare domain, for instance, compact prediction regions are essential since they enable clinicians to quickly determine whether a patient is within a healthy range or at risk, and to take timely medical action.

Motivated by this challenge, this paper asks the following question: *How can we produce provably small prediction regions satisfying the marginal coverage constraint for multi-target regression tasks?* To answer this question, we develop a novel approach referred to as *Adaptive Prediction Regions (APR)*. APR is inspired by the principle behind localized CP for single-target Guan (2023) and extends it to multi-target settings. The key idea behind APR is to exploit the heterogeneity in the conditional distribution of output given input to use a test-input conditioned quantile threshold to construct valid and small prediction regions. The effectiveness of this general idea depends on the specific localization mechanism which has not received attention. Additionally, to the best of our knowledge, localized CP method hasn't been empirically tested on real-world applications in both single-target and multi-target settings. In our work, we specify and empirically evaluate multiple instantiations of localization to address this gap in the CP literature.

To achieve this with guaranteed marginal coverage, particularly when using input-dependent weighting (APR-W), APR utilizes an $\tilde{\alpha}$-level adjustment Guan (2023) which is critical for restoring the validity of the localized quantiles. In contrast, existing multi-target CP methods such as DQR and its variants employ a uniform quantile threshold for all test inputs. We prove that APR achieves distribution-free and model-agnostic (invariant to the choice of the underlying multi-target regression method) marginal coverage guarantee. We also prove that under mild conditions on the quantiles, APR produces small prediction regions when compared to multi-target CP methods based on a uniform quantile threshold for all test inputs. Our comprehensive experiments on several real-world datasets demonstrate that APR produces significantly smaller prediction regions (by up to 85.51% reduction) compared to state-of-the-art methods, and the results validate our theory.

**Contributions.** The key contribution of this paper is the development, theoretical analysis, and evaluation of the *Adaptive Prediction Regions (APR)* algorithm for multi-target regression tasks.

Specific contributions include:

- Development of the APR algorithm, which constructs valid and small prediction regions based on the idea of test input-conditioned quantile threshold: extending the framework of localized CP Guan (2023) to multi-target regression, including the $\tilde{\alpha}$-level adjustment necessary for maintaining the marginal coverage guarantee in the weighted localized setting.

- Theoretical analysis to show that APR achieves coverage guarantee and produces smaller prediction regions compared to using uniform threshold for all test inputs.

- Localized CP is developed for single-target setting and analyzed primarily at a theoretical level. Studying effective localization schemes and empirical evaluation on real-world applications has received less attention. Therefore, APR extends this idea to the multi-target setting, specifies multiple approaches for localization, and is validated on several real-world datasets.

- Empirical evaluation on diverse real-world datasets to demonstrate the efficacy of APR over state-of-the-art baseline methods. Our code is available in the following anonymous GitHub repository `https://anonymous.4open.science/r/apr-4C4C/` for review purposes.

## 2 Background and Problem Setup

**Notations.** Let $\mathcal{D}_{\text{tr}} = \{(X_i, Y_i)\}_{i=1}^n$ be a training dataset with $n$ samples, where $X \in \mathcal{X} \subseteq \mathbb{R}^p$ and $Y \in \mathcal{Y} \subseteq \mathbb{R}^d$ are the input feature vector and output response vector defined on the input space $\mathcal{X}$ and output space $\mathcal{Y}$, respectively. We assume that all input-output pairs are independently drawn from an underlying distribution $\mathcal{P}$, i.e., $(X, Y) \sim \mathcal{P}$. Let $\mathcal{D}_{\text{cal}} = \{(X_i, Y_i)\}_{i=n+1}^{n+m}$ be a calibration data set with $m$ samples and $X_{\text{test}}$ be a test input feature vector with its corresponding response vector $Y_{\text{test}}$. Suppose $R_{\mathcal{Y}}(X) \subseteq \mathcal{Y}$ is a mapping to generate a region in output space $\mathcal{Y}$ given an input $X$.

Our goal is to construct trustworthy uncertainty regions (aka prediction regions) for multi-target regression tasks illustrated in Figure 1, so that they satisfy a conformal coverage guarantee. Specifically, we say a region-generating process $R_{\mathcal{Y}}(X)$ guarantees $(1 - \alpha)$ coverage if the following inequality holds:

$$\mathbb{P}_{(X_{\text{test}}, Y_{\text{test}}) \sim \mathcal{P}}\{Y_{\text{test}} \in R_{\mathcal{Y}}(X_{\text{test}})\} \geq 1 - \alpha. \tag{1}$$

Throughout this paper, we omit the subscript $(X, Y) \sim \mathcal{P}$ of the probability $\mathbb{P}$ about where the randomness comes from, unless otherwise specified.

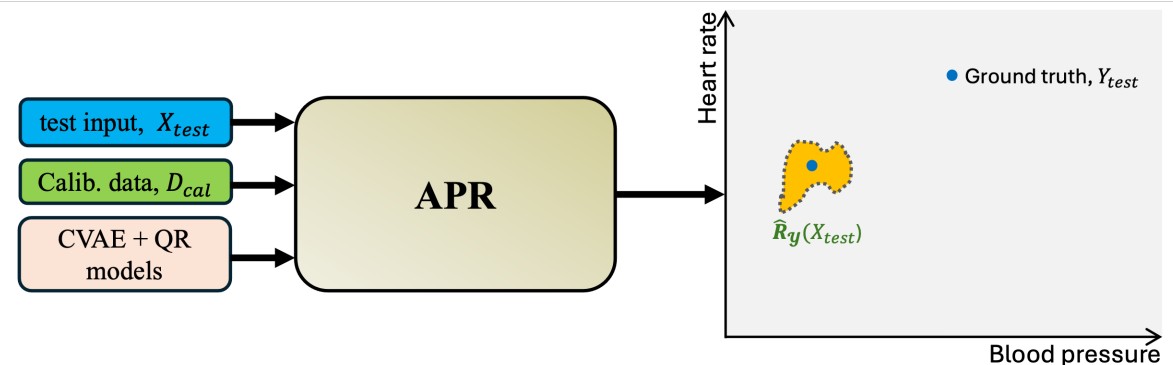

Figure 1: Illustration of the APR framework for constructing small prediction regions for a health application with two target variables (heart rate and blood pressure). Given a test input $X_{\text{test}}$, a calibration dataset $D_{cal}$, and pre-trained conditional variational autoencoder (CVAE) and multi-target quantile regression (QR) models, APR generates a compact prediction region ($\hat{R}_{\mathcal{Y}}(X_{\text{test}})$, shown in orange color) that is likely to contain the true target output (shown in blue) with a marginal coverage probability of $1 - \alpha$ (say 95%).

**Conformal Prediction** is a general framework to provide rigorous guarantees for coverage in regression and classification tasks Vovk et al. (2005); Romano et al. (2019; 2020); Gibbs et al. (2023); Tibshirani et al. (2019). CP typically relies on a *non-conformity* scoring function which measures how different a data sample is from existing ones Vovk et al. (2005). For example, in single-target regression tasks, the absolute residual $|\hat{y} - y|$ due to a regression model is a commonly used definition for the non-conformity scoring function Romano et al. (2019), where $\hat{y}$ and $y$ denote the predicted and true output, respectively. Moreover, the underlying regression model is trained and fixed during both calibration and testing stages.

Let $V : \mathcal{X} \times \mathcal{Y} \to \mathbb{R}$ denote a non-conformity scoring function.

For simplicity, we denote the non-conformity score for calibration sample $(X_j, Y_j) \in \mathcal{D}_{cal}$ by $V_j := V(X_j, Y_j)$.

Given a user-specified mis-coverage parameter $\alpha$, CP methods typically compute an empirical quantile on the calibration dataset as follows:

$$\hat{Q}(\alpha) = \inf \left\{ \tau : \frac{1}{m} \sum_{j \in \mathcal{D}_{cal}} \mathbb{1}\{V_j \leq \tau\} \geq 1 - \alpha \right\}. \tag{2}$$

For a test input $X_{\text{test}}$, we use this quantile as a threshold to selectively add candidate output responses into the prediction set:

$$\hat{\mathcal{C}}(X_{\text{test}}) = \{y \in \mathcal{Y} : V(X_{\text{test}}, y) \leq \hat{Q}(\alpha)\}.$$

It is a well-known result that if calibration data samples in $\mathcal{D}_{\text{cal}}$ and $(X_{\text{test}}, Y_{\text{test}})$ are exchangeable, then this CP procedure guarantees a marginal coverage Vovk et al. (2005):

$$\mathbb{P}\{Y_{\text{test}} \in \hat{\mathcal{C}}(X_{\text{test}})\} \geq 1 - \alpha.$$

The key difference between the above general coverage result and that in the multi-target regression setting in (1) is how the prediction set is constructed. In our problem setting, the region-generating function $R_{\mathcal{Y}}(X)$ builds the prediction region in the multi-dimensional output space (generalization of prediction interval in the single-target regression tasks). This is a significant challenge because the coverage in high-dimensional output space can be unnecessarily statistically inefficient, i.e., producing very large prediction regions to cover the true multi-target response.

**Multi-target CP algorithm.** We propose a wrapper-based solution that can use any existing multi-target method. Since we implemented our solution on top of the SOTA Feldman et al. (2023), we provide its key algorithmic steps for the sake of completeness.

SOTA begins by training a conditional variational autoencoder (CVAE) on the training dataset $\mathcal{D}_{\text{tr}}$. Specifically, we denote the CVAE by $(\mathcal{E}, \mathcal{D})$, where $\mathcal{E}$ and $\mathcal{D}$ are the encoder and decoder, respectively. Ideally, CVAE aims to fit the data to complete a two-way transform, by which it can reconstruct the conditional distribution $P(Y|X)$. The first transform is from $\mathcal{Y}$ to a latent space $\mathcal{Z} \subseteq \mathbb{R}^r$ by the encoder, i.e., to a transformed latent data point $Z_y = E(Y; X = x)$, where $r$ is the dimensionality of the latent space and can be tuned as a hyper-parameter in practice. The ideal case is that all possible latent data points are drawn from a standard Normal distribution $Z_y \sim \mathcal{N}(0, 1)$. The second transform is from the latent space $\mathcal{Z}$ to the original response space $\mathcal{Y}$ by the decoder $\mathcal{D}(Z_y; X = x) = \hat{Y}$.

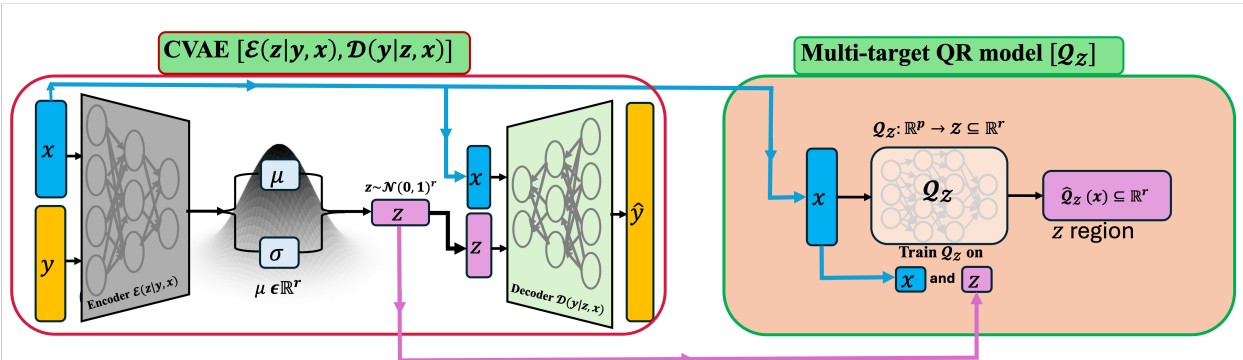

Figure 2: Overview of conditional variational autoencoder (CVAE) and multi-target quantile regression (QR) models training. The CVAE (on the left) and multi-target QR (on the right) models are trained on the training dataset to derive the encoder for mapping to the latent space $\mathcal{Z}$ and the decoder for reconstructing the original space $\mathcal{Y}$. The multi-target QR model is trained on the input ($x$) and latent target ($z$) to generate the base prediction region in the latent space $\mathcal{Z}$.

The goal of the encoder-decoder structure is to ensure that the reconstruction $\hat{Y}$ is equivalent to the true response $Y$ in distribution, i.e., $\mathcal{D}(Z_y; X = x) \stackrel{d}{=} Y|X = x$.

Once the CVAE $(\mathcal{E}, \mathcal{D})$ is trained, proceed to train a standard directional quantile regression (DQR) in the latent space $\mathcal{Z}$. For an input $X$, this creates a convex region in $\mathcal{Z}$ (since DQR only generates convex regions), denoted by $R_{\mathcal{Z}}(X)$, which is then transformed using the decoder $\mathcal{D}$ to $\mathcal{Y}$ space. Particularly, we denote the

region in $\mathcal{Y}$ that is transformed from the latent space by $R_{\mathcal{Y}}(X) = \mathcal{D}(R_{\mathcal{Z}}(X))$, which serves as a base region in $\mathcal{Y}$.

DQR either provides over- or under-coverage in $\mathcal{Y}$, so it needs further calibration on (i) whether the coverage achieved by $R_{\mathcal{Y}}(X)$ is too large or too small, and (ii) how much it needs to adjust (shrink if too large, or expand if too small) the prediction region $R_{\mathcal{Y}}(X)$ in $\mathcal{Y}$ space. In the case of under-coverage, it uses the following non-conformity scoring function, calibration step, and region-generating process:

$$V_j^+ = \min_{a \in R_{\mathcal{Y}}(X_j)} \text{dist}(a, Y_j), \forall j \in \mathcal{D}_{\text{cal}},$$

$$\Rightarrow \ \gamma^+ = \min \left\{ \tau : \frac{1}{m} \sum_{j=n+1}^{n+m} \mathbb{1}[V_j^+ \leq \tau] \geq 1 - \alpha \right\},$$

$$\Rightarrow \ R_{\mathcal{Y}}^+(X_{\text{test}}) = \left\{ y \in \mathcal{Y} : \min_{a \in R_{\mathcal{Y}}(X_{\text{test}})} \text{dist}(a, y) \leq \gamma^+ \right\} \tag{3}$$

In the case of over-coverage, the non-conformity scoring function, calibration step, and prediction region can be defined as follows:

$$V_j^- = \min_{a \in R_{\mathcal{Y}}^c(X_j)} \text{dist}(a, Y_j), \forall j \in \mathcal{D}_{\text{cal}},$$

$$\Rightarrow \ \gamma^- = \min \left\{ \tau : \frac{1}{m} \sum_{j=n+1}^{n+m} \mathbb{1}[V_j^- \leq \tau] \leq \alpha \right\},$$

$$\Rightarrow \ R_{\mathcal{Y}}^-(X_{\text{test}}) = \left\{ y \in \mathcal{Y} : \min_{a \in R_{\mathcal{Y}}(X_{\text{test}})} \text{dist}(a, y) \geq \gamma^- \right\},$$

where the $R_{\mathcal{Y}}^c(X_j) = \mathcal{Y} \backslash R_{\mathcal{Y}}(X_j)$ denotes the region that is not included by the quantile region $R_{\mathcal{Y}}(X_j)$. After either calibration step, the coverage in (1) is guaranteed to hold.

However, the calibration in SOTA does not consider the heterogeneity in the conditional probability distribution $P(Y|X)$. This is reflected in determining the quantile $\gamma^+$ and $\gamma^-$, both of which are defined in the marginal sense and are not adaptive to different realizations of test input $X_{\text{test}}$. The challenge of heterogeneous distribution $P(Y|X)$ is increasingly more important in the recent CP literature, especially when different kinds of conditional coverage notions have been proposed and investigated Gibbs et al. (2023); Vovk (2012); Ding et al. (2023). The main challenge to reduce the size of prediction regions is figuring out an algorithmic principle to capture the heterogeneity in the conditional distribution $P(Y|X)$ for multi-target conformal calibration.

**Localized Conformal Prediction.** While standard CP employs a single, global quantile threshold $\hat{Q}(\alpha)$ for all test inputs $X_{\text{test}}$, localized CP aims to compute a test input-specific quantile $Q(X_{\text{test}})$. One method to achieve this is to use weighted non-conformity scores. For a test input $X_{\text{test}}$, a localized quantile $\hat{Q}(X_{\text{test}}, \gamma)$ for a candidate confidence level $\gamma$ is defined as:

$$\hat{Q}(X_{\text{test}}, \gamma) = \min \left\{ \tau : \sum_{i \in \mathcal{D}_{\text{cal}}} w_i(X_{\text{test}}) \mathbb{1}\{V_i \leq \tau\} \geq \gamma \right\}, \tag{4}$$

where $w_i(X_{\text{test}}) \geq 0$ are weights such that $\sum_{i \in \mathcal{D}_{\text{cal}}} w_i(X_{\text{test}}) = 1$, and $V_i$ are the non-conformity scores. A common issue with using $1 - \alpha$ directly as $\gamma$ is that it does not guarantee the marginal coverage ($\mathbb{P}(Y_{\text{test}} \in \hat{\mathcal{C}}(X_{\text{test}})) \geq 1 - \alpha$) required by classic CP. To restore the marginal coverage guarantee in the weighted setting, Guan (2023) proposed learning a corrected confidence level $\tilde{\alpha}$ from the calibration set.

Specifically, for each calibration point $X_i$, let $\hat{q}_{X_i}(\gamma)$ be its localized quantile at level $\gamma$. Let $\Gamma$ be the set of all cumulative weight values attainable from the weighted CDFs. The data-driven global correction level $\tilde{\alpha}$ is computed as:

$$\tilde{\alpha} = \min_{\gamma \in \Gamma} \left\{ \gamma : \frac{1}{m} \sum_{i \in \mathcal{D}_{cal}} \mathbf{1}[V_i \leq \hat{q}_{X_i}(\gamma)] \geq 1 - \alpha \right\}. \tag{5}$$

The prediction region is then constructed using the localized quantile at level $\tilde{\alpha}$ for the test point $X_{\text{test}}$: $\hat{\mathcal{C}}(X_{\text{test}}) = \{y \in \mathcal{Y} : V(X_{\text{test}}, y) \leq \hat{Q}(X_{\text{test}}, \tilde{\alpha})\}$. This $\tilde{\alpha}$-correction ensures the finite-sample marginal coverage guarantee.

*The goal of this paper is to develop an adaptive multi-target CP algorithm that is statistically efficient to produce small prediction regions to guarantee the target marginal coverage.*

## 3 Related Work

This section summarizes the related work on conformal prediction for regression tasks. Most of the existing CP work focuses on the simpler setting of single-task regression, and there is relatively little work on CP for the multi-target setting.

**CP for single-target regression.** Conformal prediction Vovk et al. (2005); Shafer & Vovk (2008); Angelopoulos & Bates (2021); Angelopoulos et al. (2023) leverages the assumption of data exchangeability to generate prediction intervals with guaranteed coverage levels for single-target regression tasks. The standard CP approach employs the distance to the conditional mean as the conformity scoring function for calibration. Conformalized quantile regression Romano et al. (2019) integrates CP with quantile regression Regression (2017); Romano et al. (2019); Koenker & Bassett Jr (1978) estimates to construct prediction intervals. Recent work Guan (2019); Lin et al. (2021); Guan (2023) has focused on improving the calibration process to reduce the size of prediction intervals without any theoretical guarantees. To reduce the size of prediction intervals when the output distribution is complex, a recent method Guha et al. (2024) considers a reduction from regression to classification and leverages recent advances in CP for classification Angelopoulos et al. (2020); Stutz et al. (2021); Huang et al. (2023); Ding et al. (2023). However, this approach is inherently limited to single-target regression and cannot be extended to multi-target regression due to the intricate nature of its multi-dimensional continuous target space. Furthermore, no existing work has explored CP in the context of joint multi-target classification.

**CP for multi-target regression.** A naive extension of single-target CP to the multi-target setting is by independently constructing prediction intervals for each output variable, which often results in overly conservative prediction regions Feldman et al. (2023). Extending CP to the multi-target regression setting poses significant challenges. There is relatively less work in this direction and no theoretical work on analyzing the size of prediction regions. Copula-based CP Messoudi et al. (2021) leverages copulas to provide valid coverage guarantees and reliable multi-target regions. However, it produces regions that are hyper-rectangular shaped, which are typically very large and difficult to interpret. Recent works Feldman et al. (2023); Dheur et al. (2025) used recent advances in representation learning to create smaller and arbitrarily shaped prediction regions that guarantee the desired coverage. It builds on the concept of directional quantile regression (DQR) Boček & Šiman (2017); Charlier et al. (2020) by mapping the target variable to a latent convex space, constructing quantile regions in the latent space using DQR, mapping the regions back to the original output space, and then calibrating the regions for coverage using the calibration set. However, this method constructs relatively large prediction regions, which are not useful in real-world applications because it uses a uniform quantile threshold for all testing inputs.

**Probabilistic CP with approximate conditional validity.** Plassier et al. Plassier et al. (2025) propose *probabilistic* conformal prediction sets that combine conformal inference with an estimate of the conditional distribution $P_{Y|X}$, and derive non-asymptotic guarantees for *approximate conditional validity* whose tightness explicitly depends on the conditional distribution estimation error (e.g., via discrepancies such as total variation). Our work is complementary in both goal and mechanism. While probabilistic CP constructs sets by thresholding probability mass under $\widehat{P}_{Y|X=x}$ (e.g., HPD/level-set type regions) to improve conditional behavior when accurate distribution estimation is available, we focus on *multi-target regression* and ask: given a strong predictive/generative model, how can we obtain *tighter* regions while preserving the standard distribution-free *finite-sample marginal coverage* guarantee? APR answers this by *localized calibration of nonconformity scores* using neighborhood-based weighting in either the input space or a learned representation (e.g., the CVAE latent space), together with the corrected global confidence level $\tilde{\alpha}$ to maintain

marginal validity under exchangeability. Consequently, APR can yield efficiency gains even when $\widehat{P}_{Y|X}$ is not accurate enough to support conditional-coverage bounds, since our validity guarantee does not rely on conditional density accuracy and the learned model affects efficiency rather than validity.

**Theoretical comparison.** In summary, Plassier et al. Plassier et al. (2025) obtain approximate conditional validity with bounds controlled by distribution-estimation error, whereas APR guarantees finite-sample marginal validity (via $\tilde{\alpha}$) and pursues efficiency through localized score calibration in input/latent space.

# 4 Adaptive Prediction Regions Algorithm

In this section, we describe our proposed algorithm, *Adaptive Prediction Regions (APR)*, in detail. Unlike the Naive and other multi-target CP methods, which apply a uniform quantile threshold across all test inputs to construct prediction regions, APR introduces a more adaptive approach. Specifically, APR utilizes a test-input-conditioned quantile threshold to create more efficient (i.e., smaller) prediction regions. Real-world problems often involve conditional distributions $P(Y|X)$ that are inherently heterogeneous. APR leverages this heterogeneity by defining a non-uniform quantile threshold that adapts uniquely to each test input. This adaptive threshold is determined based on the top-$k$ weighted subset of calibration inputs that lie within a certain radius around the test input $X_{\text{test}}$, resulting in the construction of more adaptive prediction regions that better correspond to the true conditional distribution $P(Y|X)$.

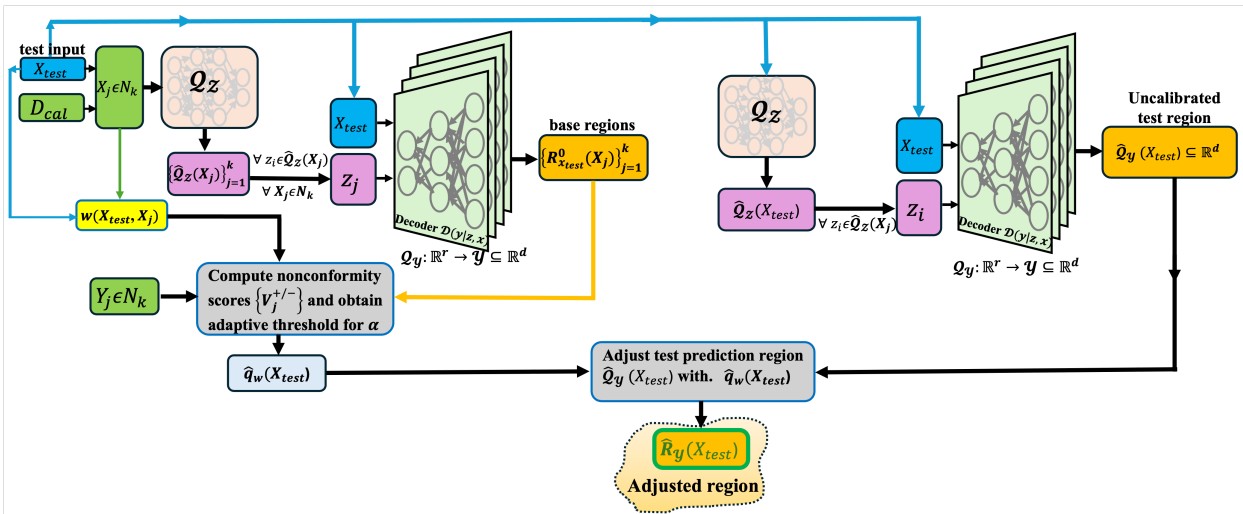

Figure 3: High-level overview of the APR algorithm illustrating the calibration and inference phases. The multi-target quantile regression model produces the initial (uncalibrated) prediction region, while the CVAE decoder maps the latent space back to the original target output space. During calibration and inference, the calibration dataset and weighting function are employed to construct a refined, smaller calibrated prediction region ($\hat{R}_{\mathcal{Y}}(X_{\text{test}})$) for the given test input $X_{\text{test}}$.

Following the procedure in Figure 2, we fit a CVAE, which comprises encoder $E(\cdot)$ and decoder $\mathcal{D}(\cdot)$, on $\mathcal{D}_{\text{tr}}$. The trained CVAE transforms the target vector $Y$ into an $r$-dimensional standard normal distribution $Z$. The transformation of CVAE ensures that $Z_i$ represents the expectation of $Y_i|X_i$. We then train a DQR model $Q_{\mathcal{Z}}$ on $\{(X_i, Z_i)\}_{i=1}^n : i \in \mathcal{D}_{tr}$ in the latent space, such that $Q_{\mathcal{Z}} : \mathbb{R}^p \to \mathcal{Z} \subseteq \mathbb{R}^r$ which constructs the base prediction region $Q_{\mathcal{Z}}(X) \subseteq \mathbb{R}^r$. Transforming $Q_{\mathcal{Z}}(X)$ back to the original $\mathcal{Y}$ space yields $Q_{\mathcal{Y}}(X) \subseteq \mathbb{R}^d$.

### 4.1 Adaptive Quantile Threshold via $\tilde{\alpha}$ Correction

**Adaptive Calibration in APR.** APR introduces a test-input conditioned quantile threshold $\hat{q}_{X_{\text{test}}}^{\text{APR}}$ to adapt the size of the prediction region based on local data density, enabling the construction of smaller regions in heterogeneous settings. The general form of the prediction region $\hat{R}_{\mathcal{Y}}(X_{\text{test}})$ for a test input $X_{\text{test}}$ is:

$$\hat{R}_{\mathcal{Y}}(X_{\text{test}}) = \{y \in \mathcal{Y} : V(X_{\text{test}}, y) \le \hat{q}_{X_{\text{test}}}^{\text{APR}}\} \tag{6}$$

where $V(\cdot, \cdot)$ is a non-conformity score (defined below as either $V^+$ or $V^-$) and $\hat{q}_{X_{\text{test}}}^{\text{APR}}$ is the final calibrated quantile threshold.

**Localized Quantile Definition.** The test-conditional quantile $\hat{q}_{X_{\text{test}}}(\gamma)$ for a candidate confidence level $\gamma$ is computed using weighted non-conformity scores $V_j$ from the calibration set $\mathcal{D}_{\text{cal}}$:

$$\hat{q}_{X_{\text{test}}}(\gamma) = \min\left\{\tau : \sum_{j \in \mathcal{D}_{\text{cal}}} w(X_{\text{test}}, X_j)\mathbb{1}\{V_j \le \tau\} \ge \gamma\right\}, \tag{7}$$

where $w(X_{\text{test}}, X_j)$ is the weighting function (detailed below) that determines the influence of each calibration sample $X_j$ based on its proximity to $X_{\text{test}}$.

**Marginal Coverage Restoration via $\tilde{\alpha}$ Correction.** For non-uniform weighting functions, using $\gamma = 1-\alpha$ in Equation (7) only guarantees conditional coverage. To ensure the desired finite-sample **marginal coverage** $\mathbb{P}\left[Y_{\text{test}} \in \hat{R}_{\mathcal{Y}}(X_{\text{test}})\right] \ge 1-\alpha$, APR employs the $\tilde{\alpha}$-level correction established in localized conformal prediction (LCP) Guan (2023). The corrected global confidence level $\tilde{\alpha}$ (where $\tilde{\alpha} \le 1 - \alpha$) is calculated using the calibration set $\mathcal{D}_{\text{cal}}$:

$$\tilde{\alpha} = \min_{\gamma \in \Gamma}\left\{\gamma : \frac{1}{m}\sum_{i \in \mathcal{D}_{cal}} \mathbf{1}[V_i \le \hat{q}_{X_i}(\gamma)] \ge 1 - \alpha\right\}, \tag{8}$$

where $\Gamma$ is the set of attainable cumulative weight values from the localized weighted CDFs $\{\widehat{F}_{X_i}\}_{i \in \mathcal{D}_{cal}}$. The final calibrated threshold for the test input is then set as $\hat{q}_{X_{\text{test}}}^{\text{APR}} = \hat{q}_{X_{\text{test}}}(\tilde{\alpha})$.

**Weighting Schemes (APR-U and APR-W).** APR primarily uses $k$-Nearest Neighbor ($k$-NN) based localization. Let $N_k(X_{\text{test}})$ be the set of $k$ nearest neighbors of $X_{\text{test}}$ in $\mathcal{D}_{\text{cal}}$, where $\text{dist}(\cdot, \cdot)$ is a user-chosen metric on the representation vectors used for neighborhood search. In all experiments in this paper, we use the standard Euclidean ($L_2$) distance after feature standardization to form $N_k(X_{\text{test}})$. For APR-W, we use inverse-distance weights within $N_k(X_{\text{test}})$, i.e., $w(X_{\text{test}}, X_j) \propto 1/(\text{dist}(X_{\text{test}}, X_j) + \varepsilon)$ with a small $\varepsilon > 0$.

$$N_k(X_{\text{test}}) = \left\{X_j \in \mathcal{D}_{\text{cal}} : \sum_{X_k \in \mathcal{D}_{\text{cal}}} \mathbb{1}[\text{dist}(X_{\text{test}}, X_k) \le \text{dist}(X_{\text{test}}, X_j)] \le k\right\}$$

There are several ways of defining the weighting function $w(X_{\text{test}}, X_j)$:

(i) **Standard-uniform weights over $\mathcal{D}_{\text{cal}}$:**

$$w(X_{\text{test}}, X_j) = 1/m. \tag{9}$$

This reduces the adaptive calibration strategy of APR back to the standard calibration that is not adaptive to the realization of test input $X_{\text{test}}$.

(ii) **APR-U** (Uniform $k$-NN): Uniform weights are assigned to the $k$ neighbors, which makes the localized weighted CDF depend only on the $k$ neighbor scores; however, because the neighborhood depends on the realized test input, we still use the corrected global level $\tilde{\alpha}$ (Eq. 8) to guarantee finite-sample *marginal* coverage.

$$w(X_{\text{test}}, X_j) = 1/k \cdot \mathbb{1}[X_j \in N_k(X_{\text{test}})] \tag{10}$$

(iii) **APR-W** (Weighted $k$-NN): Inverse-distance weights are assigned to the $k$ neighbors. This non-uniform weighting scheme is more adaptive but necessitates the full $\tilde{\alpha}$ computation (Equation 8) to guarantee marginal coverage.

$$w(X_{\text{test}}, X_j) = \frac{1/\text{dist}(X_{\text{test}}, X_j)}{\sum_{k \in N_k(X_{\text{test}})} 1/\text{dist}(X_{\text{test}}, X_k)} \cdot \mathbb{1}[X_j \in N_k(X_{\text{test}})] \tag{11}$$

(iv) **Ball-based Localizer**:

$$w(X_{\text{test}}, X_j) = \frac{\mathbb{1}[\phi(X_j) \in B(\phi(X_{\text{test}}))]}{\sum_{k \in \mathcal{D}_{cal}} \mathbb{1}[\phi(X_k) \in B(\phi(X_{\text{test}}))]} \tag{12}$$

where $\phi(X)$ is a feature mapping and $B(\cdot)$ is a Euclidean ball.

In this paper, we focus on the $k$-NN weighting functions: APR-U (Equation 10) and APR-W (Equation 11).

**Initial Base Region.** APR adapts a two-sided calibration approach to handle heterogeneity in the underlying predictor. This process starts by defining an initial base region $\mathbf{R}^{\mathbf{0}}_{\mathbf{X}_{\text{test}}}(X_j)$ for each calibration input $X_j \in N_k(X_{\text{test}})$ based on its uncalibrated region $Q_{\mathcal{Y}}(X_j)$ and a fixed initialization quantile $\hat{q}^{init}_{X_{\text{test}}}$:

$$\mathbf{R}^{\mathbf{0}}_{\mathbf{X}_{\text{test}}}(X_j) = \left\{ y \in \mathbb{R}^d : \min_{y_{in} \in Q_{\mathcal{Y}}(X_j)} \text{dist}(y_{in}, y) \leq \hat{q}^{init}_{X_{\text{test}}} \right\}, \tag{13}$$

where $Q_{\mathcal{Y}}(X_j) = \mathcal{D}(Q_{\mathcal{Z}}(X_j))$ is the uncalibrated region projected back to $\mathcal{Y}$. The $\hat{q}^{init}_{X_{\text{test}}}$ is an arbitrary initialization quantile (e.g., the $(1-\alpha)$ quantile of the distance between near points in $Q_{\mathcal{Y}}(X_{\text{test}})$) and $\text{dist}(\cdot)$ is the $L_2$ distance.

The initial coverage rate ($\mathbf{cov_{init}}$) for this base region is calculated over the $k$-NN set:

$$\mathbf{cov_{init}} = \frac{1}{k} \sum_{j \in N_k(X_{\text{test}})} \mathbb{1}[Y_j \in \mathbf{R}^{\mathbf{0}}_{\mathbf{X}_{\text{test}}}(X_j)]. \tag{14}$$

**Calibration via Score Selection.** Based on $\mathbf{cov_{init}}$, APR selects a non-conformity score ($V^+$ or $V^-$) and uses the $\tilde{\alpha}$ method to compute the final calibrated threshold $\hat{q}^{\text{APR}}_{X_{\text{test}}}$ (Equation 8 applied to the chosen score set).

**Case (i): Under-Coverage ($\mathbf{cov_{init}} \leq 1 - \alpha$).** If the desired coverage is not achieved, we use the inward-distance score $V^+$ to expand the region. $V^+$ measures the distance from the true target $Y_j$ to the closest point in the initial region $Q_{\mathcal{Y}}(X_j)$.

$$V^+_j = \min_{y_{in} \in Q_{\mathcal{Y}}(X_j)} \text{dist}(y_{in}, Y_j), \forall j \in N_k(X_{\text{test}}). \tag{15}$$

The final calibrated prediction region $\hat{R}_{\mathcal{Y}}(X_{\text{test}})$ is constructed using the threshold $\hat{q}^{\text{APR}}_{X_{\text{test}}} = \hat{q}_{X_{\text{test}}}(\tilde{\alpha})$ computed on the set of $\{V^+_j\}$ scores:

$$\hat{R}_{\mathcal{Y}}(X_{\text{test}}) = \left\{ y \in \mathbb{R}^d : \min_{y_{in} \in Q_{\mathcal{Y}}(X_{\text{test}})} \text{dist}(y_{in}, y) \leq \hat{q}^{\text{APR}}_{X_{\text{test}}} \right\}. \tag{16}$$

This $\hat{q}^{\text{APR}}_{X_{\text{test}}}$ is the localized, $\tilde{\alpha}$-corrected version of the uncorrected $\hat{q}_w(X_{\text{test}})$ from the initial adaptive calibration idea.

**Case (ii): Over-Coverage ($\mathbf{cov_{init}} > 1 - \alpha$).** If the initial region over-covers, we use the outward-distance score $V^-$ to shrink the region. $V^-$ measures the distance from the true target $Y_j$ to the closest input in the **complement** region $Q^c_{\mathcal{Y}}(X_j)$, effectively calibrating the boundary of the region.

$$V^-_j = \min_{y_{in} \in Q^c_{\mathcal{Y}}(X_j)} \text{dist}(y_{in}, Y_j), \forall j \in N_k(X_{\text{test}}), \tag{17}$$

where $Q_{\mathcal{Y}}^c(X_j)$ is the complement of $Q_{\mathcal{Y}}(X_j)$. The final calibrated prediction region $\hat{R}_{\mathcal{Y}}(X_{\text{test}})$ is constructed using the threshold $\hat{q}_{X_{\text{test}}}^{\text{APR}} = \hat{q}_{X_{\text{test}}}(\tilde{\alpha})$ computed on the set of $\{V_j^-\}$ scores:

$$\hat{R}_{\mathcal{Y}}(X_{\text{test}}) = \left\{ y \in \mathbb{R}^d : \min_{y_{in} \in Q_{\mathcal{Y}}(X_{\text{test}})} \text{dist}(y_{in}, y) \le \hat{q}_{X_{\text{test}}}^{\text{APR}} \right\}. \tag{18}$$

The key steps of the proposed APR algorithm are summarized in Algorithm 1 and illustrated in Figures 2 and 3, offering a general overview.

---

**Algorithm 1** Adaptive Prediction Regions (APR)

---

1: **Input:**
   Data $\{(X_i, Y_i)\}_{i=1}^n \subseteq \mathbb{R}^p \times \mathbb{R}^d$;   Multi-target QR algorithm $Q_{\mathcal{Z}}$;
   VAE$(y : E, \mathcal{D}) = (E(z|y), \mathcal{D}(y|z))$;   Test input $X_{\text{test}}$;   error rate $\alpha \in (0, 1)$

2: **Training CVAE and Multi-target-QR:**
3: Randomly split the training data into two disjoint sets: training $(\mathcal{D}_{tr})$ and calibration $(\mathcal{D}_{cal})$.
4: Train VAE$(y : E, \mathcal{D})$ on $\mathcal{D}_{tr}$
5: Train $Q_{\mathcal{Z}}$ on $(X_i, E(z|Y_i))$, where $E(z|Y_i) = Z_i : i \in \mathcal{D}_{tr}$ and $Z_i \sim \mathcal{N}(0, 1)^r$.
   $Q_{\mathcal{Z}}$ constructs the quantile region $Q_{\mathcal{Z}}(X) \subseteq \mathbb{R}^r$ in the latent space $\mathcal{Z}$.

6: **APR calibration and Inference:**
7: Obtain $N_k(X_{\text{test}}) \subseteq \mathcal{D}_{cal}$ and define $w(X_{\text{test}}, X_j)$ according to Eq (9) (11), (10) or (12)
8: Obtain base regions $\mathbf{R}_{\mathbf{X}_{\text{test}}}^{\mathbf{0}}(X_i), i \in N_k(X_{\text{test}})$ using Eq (13) and compute $\mathbf{cov_{init}}$ using Eq (14).
9: **if** $\mathbf{cov_{init}} \le 1 - \alpha$: **then**
10:    − Compute scores $\{V_j^+\}$ from Eq (15) and construct region $\hat{R}_{\mathcal{Y}}(X_{\text{test}})$ using Eq (16)
11: **else**
12:    − Compute scores $\{V_j^-\}$ from Eq (17) and construct region $\hat{R}_{\mathcal{Y}}(X_{\text{test}})$ using Eq (18)
13: **end if**

14: **Output:** Prediction region, $\hat{R}_{\mathcal{Y}}(X_{\text{test}}) \subseteq \mathbb{R}^d$

---

### 4.2 Theoretical Analysis

In this section, we present our theoretical analysis for the coverage guarantee of APR and its improved predictive region efficiency over the baseline. Our analysis focuses on the weighting function choice of (10) in Algorithm 1, the test input-conditional calibration with the uniform weight on $k$-NN calibration samples for test input $X_{\text{test}}$. All our complete proofs can be found in Appendix A.1.

We start with the standard definition of exchangeability for *sequences* of random variables.

**Definition 1.** *A sequence of random variables $(Z_1, \ldots, Z_n)$ is exchangeable if for any permutation $\pi$ of $\{1, \ldots, n\}$,*

$$(Z_1, \ldots, Z_n) \overset{d}{=} (Z_{\pi(1)}, \ldots, Z_{\pi(n)}).$$

*In our setting, we apply this to the sequence of examples $(X_i, Y_i)$ (calibration) together with the test example $(X_{test}, Y_{test})$.*

Importantly, selecting a neighborhood (e.g., $k$-NN) based on the realized test input can break exchangeability within the selected subset. Therefore, rather than claiming exchangeability of the $k$-NN subset, our coverage guarantee follows the localized/weighted conformal framework using the corrected global confidence level $\tilde{\alpha}$ (Eq. 8), which restores finite-sample *marginal* validity Guan (2023).

*Relationship to prior work.* Theorem 1 is an immediate corollary of Theorem 1 in Guan (2023), obtained by instantiating their localized/weighted conformal construction with our weights $w(X, \cdot)$ and the corrected level $\tilde{\alpha}$ in Equation 8.

**Theorem 1.** *(Finite-sample marginal coverage of APR with localization) Assume the calibration examples $\{(X_i, Y_i)\}_{i \in \mathcal{D}_{cal}}$ and the test example $(X_{test}, Y_{test})$ are exchangeable. Let the localized quantile $\hat{q}_X(\gamma)$ be defined via the (possibly sparse) weighting function $w(X, \cdot)$ and the corresponding weighted empirical CDF, and let $\tilde{\alpha}$ be chosen according to Eq. 8. Then the APR prediction region constructed using the calibrated threshold $\hat{q}_{X_{test}}^{\mathrm{APR}} = \hat{q}_{X_{test}}(\tilde{\alpha})$ satisfies the distribution-free finite-sample* marginal *coverage guarantee*

$$\mathbb{P}\{Y_{test} \in \hat{R}_{\mathcal{Y}}(X_{test})\} \geq 1 - \alpha.$$

**Remark 1.** The guarantee above is marginal (unconditional) coverage. The role of localization is to improve efficiency (smaller regions) under heterogeneity, while the corrected level $\tilde{\alpha}$ ensures finite-sample marginal validity under exchangeability.

A key subtlety is that the localized rule uses weights (or a $k$NN neighborhood) that depend on the realized test input $X_{\text{test}}$; hence the set of calibration scores receiving nonzero weight is not, in general, an exchangeable subset, and we do *not* assume exchangeability of the derived indicators $(A_1, \ldots, A_m, A_{\text{test}})$. Instead, our finite-sample *marginal* validity follows the localized conformal prediction argument based on *pseudo-test centers* (cf. localized CP): we choose the corrected level $\tilde{\alpha}$ to control the average pseudo-test inclusion rate across calibration centers, and exchangeability of the augmented sample yields the desired marginal coverage for the held-out test point. See Appendix A.1 for the proof.

Moreover, we highlight that the predictive region efficiency (aka area of prediction region) of different CP methods can be significantly distinct even though they guarantee the same coverage performance. Below, we show that under the concentrated condition of quantiles, our APR algorithm is more efficient in terms of the expected size of prediction regions when compared to the baseline multi-target CP method (SOTA in our study).

**Definition 2.** *(Preserving relative order of expected volume). Let $R_{\mathcal{Z}}^k(X)$ denote a kNN-localized (weighted) latent-space region generator and let $\tilde{R}_{\mathcal{Z}}(X)$ denote any (possibly non-localized) latent-space region generator; both map an input $X$ to a measurable subset of the latent space $\mathcal{Z}$. We write $|\cdot|$ for region volume (Lebesgue measure). We say that a decoder $\mathcal{D} : \mathcal{Z} \to \mathcal{Y}$ preserves the relative order of expected volume if*

$$\mathbb{E}_X\big[|R_{\mathcal{Z}}^k(X)|\big] \leq \mathbb{E}_X\big[|\tilde{R}_{\mathcal{Z}}(X)|\big] \quad \implies \quad \mathbb{E}_X\big[|\mathcal{D}(R_{\mathcal{Z}}^k(X))|\big] \leq \mathbb{E}_X\big[|\mathcal{D}(\tilde{R}_{\mathcal{Z}}(X))|\big].$$

**Remark 2.** (When does Definition 2 hold?) A sufficient condition is that the decoder is (approximately) *volume-scaling* with a nearly constant Jacobian determinant over the regions of interest, i.e., $|\det J_{\mathcal{D}}(z)| \approx c$ for $z$ inside the relevant latent regions; this is exact for affine/linear decoders (or locally affine decoders) where $|\mathcal{D}(A)| = c\,|A|$ for any measurable $A \subseteq \mathcal{Z}$, and therefore preserves the relative order of expected volumes. More generally, if $\mathcal{D}$ is a smooth injective map whose Jacobian determinant is bounded and slowly varying over the candidate regions (so that local volume distortion is controlled), then the ordering in expected latent volume is preserved in target space up to small approximation error. In our setting, we only require this property *locally* on the family of APR regions actually produced by the method (calibrated balls around $\mu_{\mathcal{Z}}(X)$), rather than globally for all measurable subsets of $\mathcal{Z}$. This is consistent with smooth decoders that do not exhibit extreme expansion/compression on these regions, and it can be sanity-checked by verifying that smaller calibrated radii in $\mathcal{Z}$ typically correspond to smaller decoded-region size in $\mathcal{Y}$ on held-out data.

The preservation of the relative order of expected volume from $\mathcal{Z}$ to $\mathcal{Y}$ allows us to analyze the predictive efficiency. Based on the ideal CVAE, the latent variable in the latent space $\mathcal{Z}$ follows the multivariate Gaussian distribution $\mathcal{N}(0, 1)^r$, which enables many statistical tools to understand how the density is distributed. We use the superscript "$k$" to explicitly indicate *kNN-localization with respect to the test input $X$*. Concretely: (i) $R_{\mathcal{Z}}^k(X)$ denotes the *kNN-based* (localized/weighted) latent-space prediction region constructed using only the $k$ nearest calibration inputs to $X$; (ii) $R_{\mathcal{Z}}(X)$ denotes the *non-localized* (global) latent-space prediction region constructed without $k$NN localization. We then define the corresponding target-space regions by decoding: $R_{\mathcal{Y}}^k(X) := \mathcal{D}\big(R_{\mathcal{Z}}^k(X)\big)$ and $R_{\mathcal{Y}}(X) := \mathcal{D}\big(R_{\mathcal{Z}}(X)\big)$. Additionally, since APR thresholds an isotropic latent score of the form $\|z - \mu_{\mathcal{Z}}(X)\|_2$ with a scalar calibrated radius, the latent regions $R_{\mathcal{Z}}^k(X)$ and $R_{\mathcal{Z}}(X)$ are (by construction) $r$-dimensional Euclidean balls in $\mathcal{Z}$.

**Assumption 1** (Local quantile shrinkage under $k$NN localization)**.** *For each input $X$, let $v^k(X)$ denote the calibrated latent-space radius produced by APR with $kNN$-localized weights, and let $v^{\mathrm{glob}}(X)$ denote the corresponding calibrated radius produced without localization (i.e., using global/uniform weights). We assume that*

$$v^k(X) \leq v^{\mathrm{glob}}(X) \quad \text{almost surely in } X.$$

*Equivalently, the localized weighted score distribution is (on average) no more dispersed than the global one, so its calibrated quantile radius is no larger.*

**Remark 3.** [Why Assumption 1 is reasonable] This assumption is reasonable because $k$NN localization conditions the calibration scores on a neighborhood of inputs that are similar to $X$. If the score distribution varies with $X$ (heteroskedasticity) but is relatively homogeneous within local neighborhoods, then restricting calibration to nearby points reduces mixing across disparate regimes, yielding a less dispersed score distribution and therefore a smaller calibrated quantile radius. A sufficient condition is that the input-conditional score dispersion is locally smaller than its global dispersion for most $X$.

**Theorem 2.** *(Improved prediction region efficiency of APR). Suppose all calibration samples $(X_i, Y_i) \in \mathcal{D}_{cal}$ and the test pair $(X, Y)$ are exchangeable. Let $R_{\mathcal{Z}}^k(X)$ be the $kNN$-localized latent-space prediction region produced by APR (i.e., constructed using weights supported on the $k$ nearest calibration inputs to $X$), and let $R_{\mathcal{Z}}(X)$ be the corresponding non-localized (global) latent-space prediction region obtained without $kNN$ localization. Define the decoded target-space regions*

$$R_{\mathcal{Y}}^k(X) := \mathcal{D}\big(R_{\mathcal{Z}}^k(X)\big) \quad \text{and} \quad R_{\mathcal{Y}}(X) := \mathcal{D}\big(R_{\mathcal{Z}}(X)\big).$$

*Assume Assumption 1 and that the decoder $\mathcal{D}$ preserves the relative order of expected volume in the sense of Definition 2. Then APR's $kNN$-localized target-space regions are no larger in expected volume than the non-localized regions, i.e.,*

$$\mathbb{E}_X\big[|R_{\mathcal{Y}}^k(X)|\big] \leq \mathbb{E}_X\big[|R_{\mathcal{Y}}(X)|\big].$$

The above result demonstrates that the uniform $k$-NN weighting function improves the predictive efficiency of uncertainty regions while ensuring that the coverage is achieved. The improvement mainly comes from the concentration of Gaussian random variables in $\mathcal{Z}$ when the CVAE is ideally learned. We report the extensive empirical results below to support the theoretical insights.

**Remark 4. Assumptions and generality of Theorem 2.** Theorem 2 is an *efficiency* result (region size) and is not required for the marginal validity guarantee in Theorem 1. Its assumptions are stylized sufficient conditions that make the comparison analytically tractable. First, the "Gaussian latent" condition is motivated by the standard VAE prior $Z \sim \mathcal{N}(0, I)$; exact Gaussianity is idealized, but the same concentration intuition extends to approximately isotropic sub-Gaussian / log-concave latent distributions where high-probability mass concentrates. Second, "ideal" training of the underlying multi-target quantile model is used to simplify the geometry of the uncalibrated region in $\mathcal{Z}$ (approximately radial / symmetric), enabling clean volume comparisons. Third, the decoder assumption (preserving relative order of expected volume) is a sufficient regularity condition. For example, if the decoder $\mathcal{D}$ is approximately bi-Lipschitz on the relevant latent region, i.e., there exist constants $0 < c \leq C$ such that

$$c\|z_1 - z_2\| \leq \|\mathcal{D}(z_1) - \mathcal{D}(z_2)\| \leq C\|z_1 - z_2\|,$$

then volumes scale within constants: $c^d|R| \leq |\mathcal{D}(R)| \leq C^d|R|$, which preserves ordering up to multiplicative factors. In practice, these assumptions are not expected to hold exactly; rather, Theorem 2 provides a principled explanation of when localization yields smaller regions (heterogeneity + concentrated representations), which we corroborate empirically.

# 5 Experiments and Results

In this section, we present the experimental evaluation of the proposed APR method, comparing it against a naive baseline and the state-of-the-art SOTA method. We discuss the results in terms of the validity and size of the prediction region area (generally interpreted as hypervolume). For simplicity, we refer to this as "*area*" throughout the paper.

| Dataset | Targets | Methods | Cov. | Relative Region Area ↓ | Reduction(%) from Naive ↑ | Reduction(%) from SOTA ↑ |
|---|---|---|---|---|---|---|
| BIWI_2 | 2 | Naive | 0.92 | 1.74 | — | — |
| | | SOTA | 0.91 | 1.18 | 32.42% | — |
| | | APR-U | 0.90 | 1.10 | 37.03% | 6.81% |
| | | APR-W | 0.89 | 1.00 | **42.61%** | **15.08%** |
| BIWI_3 | 3 | Naive | 0.93 | 3.06 | — | — |
| | | SOTA | 0.92 | 1.28 | 57.95% | — |
| | | APR-U | 0.91 | 1.17 | 61.63% | 8.76% |
| | | APR-W | 0.90 | 1.00 | **67.27%** | **22.17%** |
| Community_2 | 2 | Naive | 0.90 | 2.36 | — | — |
| | | SOTA | 0.91 | 1.07 | 54.59% | — |
| | | APR-U | 0.89 | 1.01 | 57.28% | 5.91% |
| | | APR-W | 0.89 | 1.00 | **57.57%** | **6.55%** |
| Community_3 | 3 | Naive | 0.90 | 4.99 | — | — |
| | | SOTA | 0.90 | 1.10 | 77.94% | — |
| | | APR-U | 0.91 | 1.02 | 79.65% | 7.75% |
| | | APR-W | 0.90 | 1.00 | **79.96%** | **9.18%** |
| Community_4 | 4 | Naive | 0.90 | 11.60 | — | — |
| | | SOTA | 0.91 | 1.16 | 90.02% | — |
| | | APR-U | 0.90 | 1.03 | 91.16% | 11.37% |
| | | APR-W | 0.90 | 1.00 | **91.38%** | **13.56%** |
| Bio | 2 | Naive | 0.90 | 1.17 | — | — |
| | | SOTA | 0.90 | 1.00 | **14.33%** | — |
| | | APR-U | 0.90 | 1.01 | 13.60% | -0.86% |
| | | APR-W | 0.90 | 1.01 | 13.70% | -0.74% |
| House | 2 | Naive | 0.90 | 1.18 | — | — |
| | | SOTA | 0.90 | 1.04 | 11.52% | — |
| | | APR-U | 0.89 | 1.00 | 14.91% | 3.83% |
| | | APR-W | 0.89 | 1.00 | **15.05%** | **3.98%** |
| Blog | 2 | Naive | 0.90 | 1.13 | — | — |
| | | SOTA | 0.90 | 1.20 | -5.98% | — |
| | | APR-U | 0.87 | 1.01 | 10.74% | 15.78% |
| | | APR-W | 0.87 | 1.00 | **11.73%** | **16.71%** |
| Maint._2 | 2 | Naive | 0.90 | 22.03 | — | — |
| | | SOTA | 0.99 | 6.96 | 68.41% | — |
| | | APR-U | 0.86 | 1.00 | **95.46%** | **85.63%** |
| | | APR-W | 0.95 | 1.01 | 95.42% | 85.51% |
| Maint._3 | 3 | Naive | 0.91 | 4.87e2 | — | — |
| | | SOTA | 0.98 | 1.44 | 99.70% | — |
| | | APR-U | 0.88 | 1.17 | 99.76% | 18.87% |
| | | APR-W | 0.94 | 1.00 | **99.79%** | **30.56%** |
| Maint._4 | 4 | Naive | 0.91 | 1.24e4 | — | — |
| | | SOTA | 0.98 | 1.39 | 99.99% | — |
| | | APR-U | 0.98 | 1.00 | 99.99% | 28.20% |
| | | APR-W | 0.87 | 1.00 | **99.99%** | **28.12%** |

Table 1: Coverage rates, relative region size, and reduction in region area size of APR relative to SOTA method in target space $\mathcal{Y}$ presented for eleven datasets with multiple targets at $\alpha = 0.1$. Results for each dataset are averaged over 20 experimental runs with standard errors provided. Detailed raw experimental data and standard errors are provided in Appendix A.3.

## 5.1 Experimental Setup

**Datasets.** We employed multiple real-world datasets, consistent with those used in Romano et al. (2019); Feldman et al. (2023), and additional datasets spanning three broad application domains: 1) Healthcare,

| Dataset | Targets | Methods | Cov. | Relative Region Area ↓ | Reduction(%) from Naive ↑ | Reduction(%) from SOTA ↑ |
|---|---|---|---|---|---|---|
| BIWI_2 | 2 | Naive | 0.97 | 2.60 | – | – |
| | | SOTA | 0.96 | 1.25 | 51.86% | – |
| | | APR-U | 0.94 | 1.08 | 58.42% | 13.63% |
| | | APR-W | 0.93 | 1.00 | **61.54%** | **20.10%** |
| BIWI_3 | 3 | Naive | 0.96 | 5.11 | – | – |
| | | SOTA | 0.96 | 1.24 | 75.81% | – |
| | | APR-U | 0.95 | 1.18 | 76.98% | 4.84% |
| | | APR-W | 0.94 | 1.00 | **80.43%** | **19.08%** |
| Blog | 2 | Naive | 0.95 | 1.34 | – | – |
| | | SOTA | 0.96 | 1.14 | 14.94% | – |
| | | APR-U | 0.93 | 1.01 | 24.43% | 11.16% |
| | | APR-W | 0.93 | 1.00 | **25.24%** | **12.11%** |
| Bio | 2 | Naive | 0.94 | 1.22 | – | – |
| | | SOTA | 0.95 | 1.02 | 16.74% | – |
| | | APR-U | 0.94 | 1.00 | 17.86% | 1.35% |
| | | APR-W | 0.95 | 1.00 | **17.98%** | **1.49%** |
| Community_3 | 3 | Naive | 0.96 | 6.36 | – | – |
| | | SOTA | 0.96 | 1.14 | 82.11% | – |
| | | APR-U | 0.95 | 1.02 | 84.02% | 10.70% |
| | | APR-W | 0.95 | 1.00 | **84.28%** | **12.15%** |
| Community_4 | 4 | Naive | 0.95 | 13.72 | – | – |
| | | SOTA | 0.96 | 1.12 | 91.86% | – |
| | | APR-U | 0.95 | 1.03 | 92.53% | 8.19% |
| | | APR-W | 0.95 | 1.00 | **92.71%** | **10.45%** |
| Community_2 | 2 | Naive | 0.96 | 2.82 | – | – |
| | | SOTA | 0.96 | 1.08 | 61.70% | – |
| | | APR-U | 0.95 | 1.01 | 64.26% | 6.68% |
| | | APR-W | 0.95 | 1.00 | **64.55%** | **7.42%** |
| House | 2 | Naive | 0.95 | 1.30 | – | – |
| | | SOTA | 0.95 | 1.07 | 17.58% | – |
| | | APR-U | 0.94 | 1.00 | 22.79% | 6.33% |
| | | APR-W | 0.94 | 1.00 | **22.96%** | **6.53%** |
| Maint._2 | 2 | Naive | 0.94 | 107.45 | – | – |
| | | SOTA | 0.99 | 8.92 | 91.70% | – |
| | | APR-U | 0.92 | 1.00 | 99.07% | 88.79% |
| | | APR-W | 0.97 | 1.07 | **99.00%** | **87.98%** |
| Maint._3 | 3 | Naive | 0.95 | 3275.57 | – | – |
| | | SOTA | 0.98 | 4.24 | 99.87% | – |
| | | APR-U | 0.93 | 1.00 | 99.97% | 76.41% |
| | | APR-W | 0.97 | 1.00 | **99.97%** | **76.35%** |
| Maint._4 | 4 | Naive | 0.95 | 210671.15 | – | – |
| | | SOTA | 0.97 | 10.87 | 99.99% | – |
| | | APR-U | 0.93 | 1.13 | 100.00% | 89.62% |
| | | APR-W | 0.99 | 1.00 | **100.00%** | **90.80%** |

Table 2: Coverage rates, relative region size, and reduction in region area size of APR relative to SOTA method in target space $\mathcal{Y}$ presented for eleven datasets with multiple targets at $\alpha = 0.05$. Results for each dataset are averaged over 20 experimental runs with standard errors provided. Detailed raw experimental data and standard errors are provided in Appendix A.3.

2) Social sciences, and 3) Engineering. These datasets include Communities and Crime Dataset (Communities_2, Communities_3, Communities_4 for two, three, and four targets respectively) Redmond (2002), a dataset on physicochemical properties of protein tertiary structure (Bio) Rana (2013), House Sales in King County, USA (House) hou (2015), Blog feedback (Blog) Buza (2014), and the AI4I 2020 Predictive Maintenance Datasets (Maint._2, Maint._3, and Maint._4 for two, three, and four targets respectively) mis (2020). We additionally evaluate on the BIWI Kinect Head Pose dataset Fanelli et al. (2013) (denoted BIWI_2/, BIWI_3 depending on the number of pose targets), where each example is a face image and the targets are head-pose angles. Following the experimental protocol used in our prior draft, we featurize each image by flattening it to a 307,200-dimensional vector and then reducing the dimension to 3,072 via PCA before training. This ensures that the $k$-NN localization is performed on a low-dimensional, well-conditioned representation rather than in the raw pixel space. For datasets that originally featured 1-D targets (such as Bio, House, and Blog), we adapted them to include 2-D targets, following the approach in Feldman et al. (2023), making them appropriate for multi-target regression tasks. Further details on the number of targets and training, testing, validation, and calibration samples for each dataset are provided in Appendix A.2.

**Configuration of algorithms and baselines.** We compare two variants of our proposed APR method wrapped around SOTA, namely, `APR-U` (with uniform weights) and `APR-W` (with non-uniform weights), against baseline methods including `Naive` (independent CP for each target variable) and `SOTA` Feldman et al. (2023). Unless otherwise stated, we set the desired coverage level to $(1 - \alpha) = 0.9$. We additionally evaluate $\alpha \in \{0.1, 0.05\}$ (i.e., 90% and 95% prediction regions) and report $\alpha = 0.05$ results in Appendix A.3. We focus on these two operating points because very small $\alpha$ (e.g., $\alpha < 0.01$) often yields extremely large multi-target regions that are difficult to visualize and compare meaningfully across datasets. For APR, we choose the neighborhood size $k$ via a systematic sweep over $k \in [0.3m, 0.9m]$ (where $m = |\mathcal{D}_{\text{cal}}|$) and select the value that minimizes the prediction-region area; Table 5 reports the resulting average $k$ values. We split the dataset as follows: 20% for testing, 16% for validation (used for early stopping), 12.8% for calibration, and the remaining 51.2% for training. This is achieved by first allocating 20% of the dataset to testing. Then, from the remaining data, 20% is set aside for validation, 20% of what remains after that is used for calibration, and the rest is used for training. To provide a common basis for comparison, we set the latent space $\mathcal{Z}$ for SOTA and APR to $r = 3$ and evaluated performance (coverage and prediction region area).

The Naive and SOTA methods were implemented using the official code available at `https://github.com/Shai128/mqr`. The experiments were run on a machine with Rocky Linux 8.10 (Green Obsidian) OS, an AMD EPYC 7573X 32-Core Processor, and two NVIDIA A40 GPUs (each with 46 GB of memory), using GPU Driver Version 555.42.02 and CUDA Version 12.5.

**Evaluation methodology.** We evaluate all methods using two metrics: 1) `coverage` and 2) `prediction region area`. Coverage is computed as the proportion of test samples for which the correct multi-target output is included within the predicted region. The prediction region area is calculated by discretizing the target output space $\mathcal{Y}$ into a grid and counting the number of grid points within the prediction region Feldman et al. (2023). We report coverage, region area, and the percentage reduction in region area relative to the Naive and SOTA baselines. When needed (e.g., in Figures 5–6), we normalize region areas by the smallest method so that the best method has relative area $= 1.0$. The results reported in this work average over 20 runs across all methods and datasets. Unless otherwise stated, we set the desired coverage level to $(1-\alpha) = 0.9$ and also report results for $\alpha \in \{0.1, 0.05\}$. We select the hyperparameter value $\lambda$ using validation data. To choose $k$ for the test input-conditioned quantile threshold in APR, we performed a systematic search between 30% and 90% of the calibration set, selecting the $k$ value that provides the smallest region size in the validation phase. We provide the average $k$ value (as a percentage of the calibration set) used by APR across all datasets in Appendix A.2.

## 5.2 Synthetic Data for Heterogeneity Analysis

We perform experiments on two synthetic datasets shown in Figure 8 to demonstrate the efficacy of APR in constructing smaller prediction regions when the conditional distribution $P(Y|X)$ is highly heterogeneous, particularly when the input data $X$ exhibits clustering. Since APR uses localized calibration via $k$-NN weights, it can effectively restrict the calibration set to inputs that are most likely drawn from the same

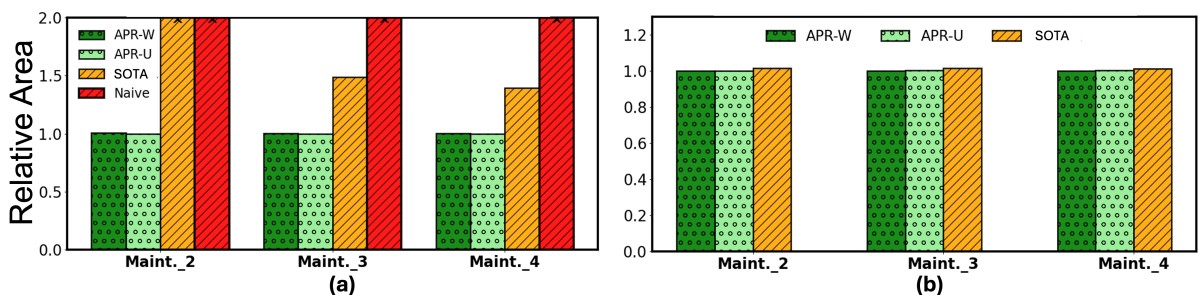

Figure 4: Relative region areas for Maintenance datasets. **a)** shows the relative region areas for APR, SOTA, and Naive methods in the target space $\mathcal{Y}$, while **b)** shows the areas for APR and SOTA methods in the latent space $\mathcal{Z}$ over 20 runs. The results show that in both $\mathcal{Y}$ and $\mathcal{Z}$ spaces, APR-based methods produce the smallest region (with relative area $= 1$), more prominent in our target space of interest $\mathcal{Y}$.

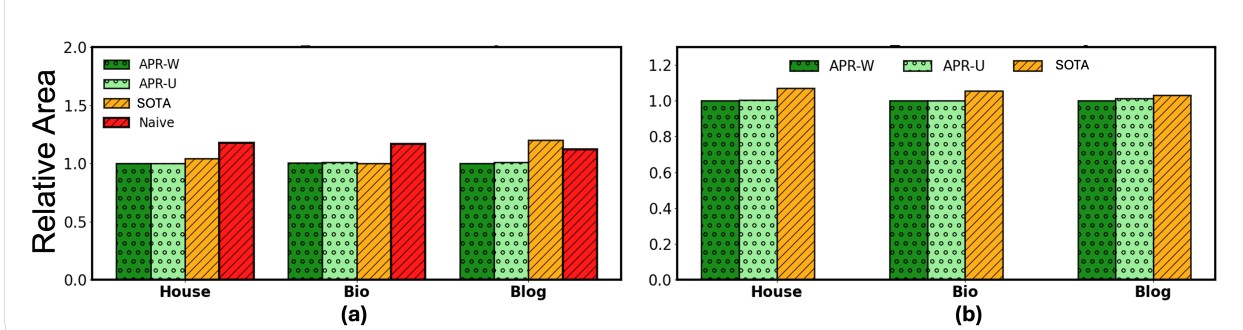

Figure 5: Relative region areas for House, Bio., and Blog datasets. **a)** shows the relative region areas for APR, SOTA, and Naive methods in the target space $\mathcal{Y}$, while **b)** shows the areas for APR and SOTA methods in the latent space $\mathcal{Z}$ over 20 runs. The results show that in both $\mathcal{Y}$ and $\mathcal{Z}$ spaces, APR-based methods generally produce the smallest region (with relative area $= 1$).

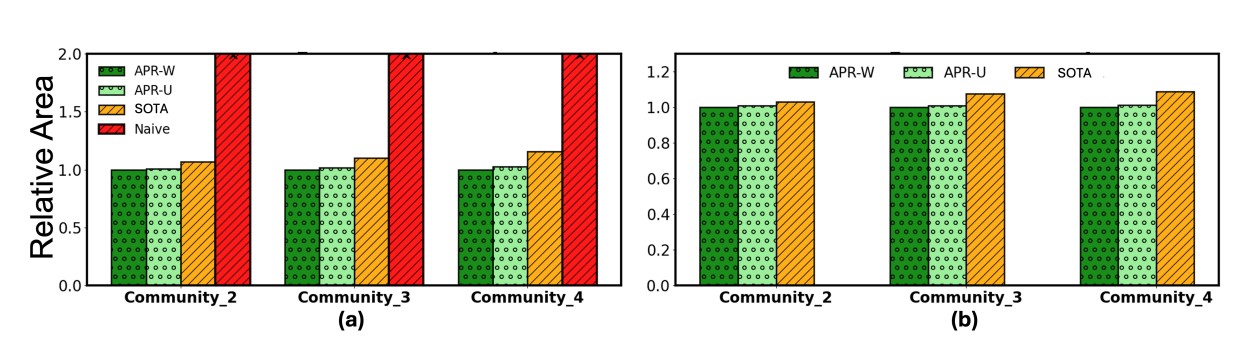

Figure 6: Relative region areas for Community and Crimes datasets. **a)** shows the relative region areas for APR, SOTA, and Naive methods in the target space $\mathcal{Y}$, while **b)** shows the areas for APR and SOTA methods in the latent space $\mathcal{Z}$ over 20 runs. The results show that in both $\mathcal{Y}$ and $\mathcal{Z}$ spaces, APR-based methods produce the smallest region (with relative area $= 1$).

| Dataset | Targets | Methods | Cov. | Region Area $\downarrow$ | Relative Region Area $\downarrow$ | Reduction(%) from SOTA $\uparrow$ |
|---|---|---|---|---|---|---|
| Community_2 | 2 | SOTA | 0.89 | 21354.32 | 1.03 | — |
| | | APR-U | 0.89 | 20943.52 | 1.01 | 1.92% |
| | | APR-W | 0.89 | 20728.20 | **1.00** | **2.93%** |
| Community_3 | 3 | SOTA | 0.92 | 19500.43 | 1.08 | — |
| | | APR-U | 0.90 | 18288.69 | 1.01 | 6.21% |
| | | APR-W | 0.90 | 18107.41 | **1.00** | **7.14%** |
| Community_4 | 4 | SOTA | 0.91 | 23378.38 | 1.09 | — |
| | | APR-U | 0.93 | 21758.72 | 1.01 | 6.93% |
| | | APR-W | 0.93 | 21504.10 | **1.00** | **8.02%** |
| Bio | 2 | SOTA | 0.91 | 18869.40 | 1.07 | — |
| | | APR-U | 0.89 | 17709.60 | 1.00 | 6.15% |
| | | APR-W | 0.89 | 17645.01 | **1.00** | **6.49%** |
| House | 2 | SOTA | 0.90 | 17013.68 | 1.06 | — |
| | | APR-U | 0.88 | 16158.43 | 1.00 | 5.03% |
| | | APR-W | 0.88 | 16130.17 | **1.00** | **5.19%** |
| Blog | 2 | SOTA | 0.89 | 18363.30 | 1.03 | — |
| | | APR-U | 0.89 | 18075.51 | 1.01 | 1.57% |
| | | APR-W | 0.88 | 17818.78 | **1.00** | **2.97%** |
| Maint._2 | 2 | SOTA | 0.95 | 26836.81 | 1.00 | — |
| | | APR-U | 0.94 | 26468.54 | 1.02 | 1.37% |
| | | APR-W | 0.94 | 26426.70 | **1.00** | **1.53%** |
| Maint._3 | 3 | SOTA | 0.95 | 19941.66 | 1.02 | — |
| | | APR-U | 0.96 | 19700.93 | 1.00 | 1.21% |
| | | APR-W | 0.96 | 19649.40 | **1.00** | **1.47%** |
| Maint._4 | 4 | SOTA | 0.93 | 12832.50 | 1.01 | — |
| | | APR-U | 0.93 | 12721.29 | 1.00 | 0.87% |
| | | APR-W | 0.93 | 12689.18 | **1.00** | **1.12%** |

Table 3: Coverage rates, relative region size, and reduction in region area size of APR relative to SOTA method in latent space $\mathcal{Z}$ presented for eleven datasets with multiple targets. Results for each dataset are averaged over 20 experimental runs, with standard errors provided.

Table 4: Final results on `clustered_close_1d` and `clustered_spread_1d` with test ratio 0.2 and calibration ratio 0.1. Reported values are averaged over 20 runs.

| Dataset | Method | Area | Rel. area | Reduction vs. SOTA (%) |
|---|---|---|---|---|
| clustered_close_1d | SOTA | 100.55 | 1.37 | — |
| | APR | 73.37 | 1.00 | 27.01 |
| clustered_spread_1d | SOTA | 274.60 | 1.66 | — |
| | APR | 165.02 | **1.00** | **39.91** |

local data generation process as the test input $X_{\text{test}}$. This dramatically reduces the prediction region size compared to standard Conformal Prediction (CP) methods that calibrate globally.

**Dataset Generation.** Both synthetic datasets feature a 1-dimensional input $X \in \mathbb{R}$ and a 2-dimensional target $Y = (Y_1, Y_2) \in \mathbb{R}^2$. The data is generated from a mixture of $K = 3$ Gaussian clusters, where the cluster index $K$ is sampled uniformly, $K \sim \text{Unif}\{1, 2, 3\}$.

For the `clustered_spread_1d` dataset, we use three well-separated clusters centered at $-3$, 0, and 3:

$$\mu_1^{(\text{spread})} = -3, \quad \mu_2^{(\text{spread})} = 0, \quad \mu_3^{(\text{spread})} = 3,$$

$$X \mid K = k \sim \mathcal{N}\big(\mu_k^{(\text{spread})}, \sigma_{\text{spread}}^2\big),$$

for a small variance $\sigma_{\text{spread}}^2 > 0$ so that the three components are clearly separated on the real line. This scenario represents strong input clustering and heterogeneity.

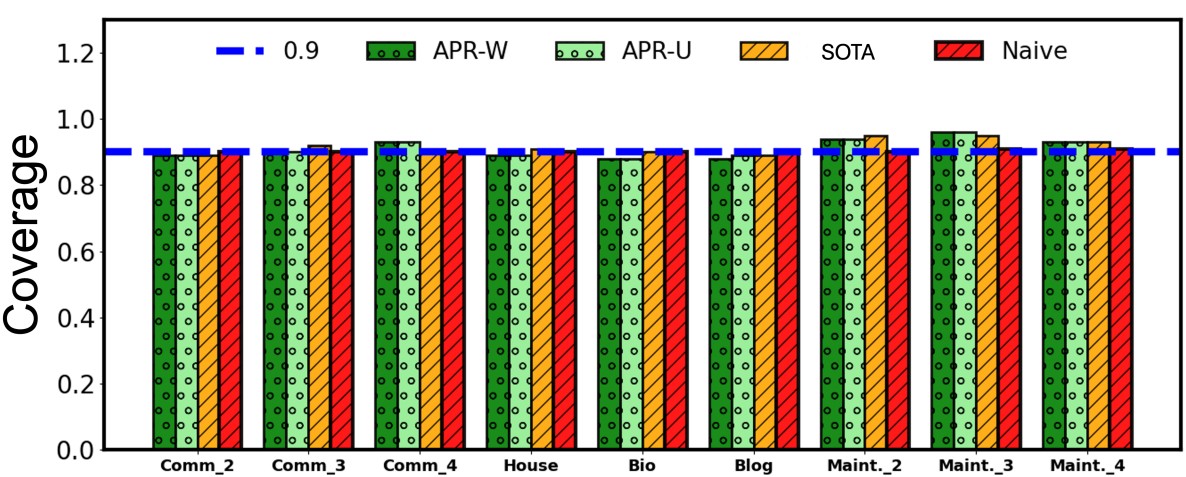

Figure 7: Empirical coverage for APR-based methods, SOTA, and Naive methods in target space $\mathcal{Y}$ (for Naive) and latent space $\mathcal{Z}$ over 20 runs. The results show that all methods generally achieve empirical coverage closer to the target level of 0.9.

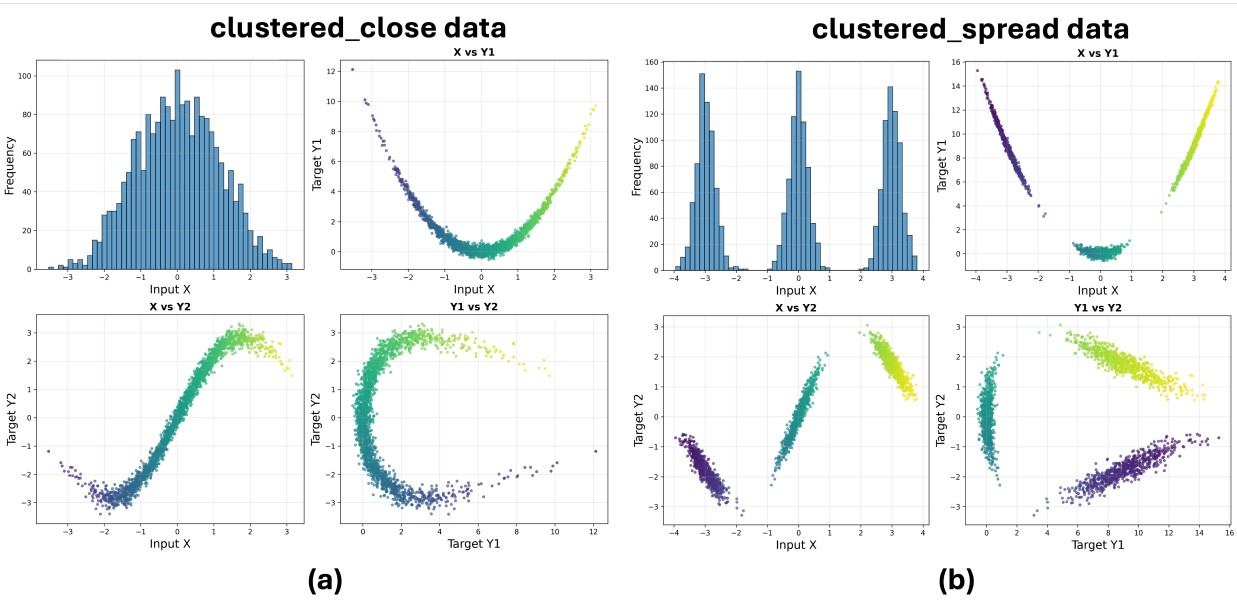

Figure 8: Synthetic clustered datasets used in our experiments. Panel (a) shows the `clustered_close_1d` data, where the one-dimensional covariate $X$ is approximately unimodal and the two targets $(Y_1, Y_2)$ vary smoothly with $X$. Panel (b) shows the `clustered_spread_1d` data, where $X$ forms three well-separated clusters and the corresponding targets form distinct, cluster-specific patterns. The clear separation of clusters in `clustered_spread_1d` highlights the strong input-dependent heterogeneity that APR exploits.

For the `clustered_close_1d` dataset, we instead place the clusters closer together, with centers at $-1$, $0$, and $1$:

$$\mu_1^{(\text{close})} = -1, \quad \mu_2^{(\text{close})} = 0, \quad \mu_3^{(\text{close})} = 1,$$

$$X \mid K = k \sim \mathcal{N}\big(\mu_k^{(\text{close})}, \sigma_{\text{close}}^2\big),$$

with $\sigma_{\text{close}}^2 > \sigma_{\text{spread}}^2$ so that the components overlap and the resulting clusters in $X$ are much less clearly separated. This represents weak input clustering.

In both synthetic datasets, the two targets are generated from smooth nonlinear functions of $X$ with independent Gaussian noise:

$$Y_1 = X + 0.5\sin(2X) + \varepsilon_1,$$

$$Y_2 = 0.5X^2 + \varepsilon_2,$$

$$\varepsilon_1, \varepsilon_2 \overset{\text{i.i.d.}}{\sim} \mathcal{N}(0, \sigma_y^2),$$

for some noise variance $\sigma_y^2 > 0$. Thus, the conditional distribution $P(Y \mid X)$ is smooth within each cluster, while the marginal distribution of $X$ provides the main source of heterogeneity in the input space.

The results for the synthetic data experiments in Table4 show that APR substantially shrinks the prediction region area relative to SOTA, with the largest gains appearing on `clustered_spread_1d` where the input clusters are well separated. In this setting, APR can focus its kernel weights on calibration inputs drawn from the same cluster as the test input, yielding an area reduction of 39.91% compared to SOTA, whereas on `clustered_close_1d` the corresponding reduction is 27.01%. These synthetic experiments therefore provide a controlled illustration of APR's main advantage: when covariates form well-separated clusters, localized calibration around each test input produces markedly tighter multi-target prediction regions without compromising coverage.

### 5.3    Results and Discussion

Our experimental results are summarized in Tables 1 and 3. Table 1 shows the coverage, relative region area, and reduction in prediction region area with respect to Naive and SOTA, the state-of-the-art multi-target CP method in the target output space $\mathcal{Y}$. Table 3 shows similar results for the best-performing variant of APR (APR-W) and SOTA in the latent space $\mathcal{Z}$. Figure 7 shows the empirical coverages obtained by the Naive, SOTA, and APR-based methods. We summarize our key experimental findings below.

**Empirical validation of the APR theory.** We make the following observations from Tables 1 and 3, and Figures 4, 5, 6, and 7. **1)** APR methods generally achieve empirical marginal coverage on all datasets, validating Theorem 1. **2)** APR's coverage distribution in Figures 7 is closer to the target coverage level (0.9), providing robust empirical support for the theoretical guarantee in Theorem 1. **3)** Results in Tables 1 and 3 demonstrate the effectiveness of APR in reducing the prediction region area over the SOTA method, which uses a uniform threshold for all test inputs. **4)** Across both target space $\mathcal{Y}$ and latent space $\mathcal{Z}$, APR constructs smaller prediction regions than the baseline for all datasets, which empirically shows that the decoder preserved the relative order of the expected volume when transforming from $\mathcal{Z}$ to $\mathcal{Y}$ space as posited in Theorem 2.

**APR-based methods vs. state-of-the-art.** We make the following observations from Tables 1 and 3. **1)** All multi-target CP methods, including APR variants and SOTA, produce smaller prediction regions when compared to the Naive baseline. This result demonstrates the importance of joint reasoning by exploiting the correlations between target variables to construct prediction regions. **2)** APR-W variant with non-uniform weights for k-NN calibration examples performs better than APR-U in most cases. This result demonstrates the importance of distance-based non-uniform weighting. **3)** While all methods approximately achieve the nominal target coverage level, APR-based methods consistently produced significantly smaller and more adaptive prediction regions. When compared to the state-of-the-art SOTA method, APR achieves a maximum reduction of 85.51% in the prediction region area in the target output space $\mathcal{Y}$. This result

demonstrates the importance of the test input-conditional quantile threshold approach in reducing prediction region sizes. Further detailed experimental results and comparisons showcasing the efficacy of APR-based methods over baseline methods are provided in Appendix A.3.

## 6 Summary and Future Work

This paper studied a provable conformal prediction approach for multi-target regression tasks called Adaptive Prediction Regions (APR). APR relies on test input-conditioned quantile threshold to generate small and valid prediction regions that adapt to each test input. Our experiments on diverse real-world datasets demonstrate that APR significantly reduces the size of prediction regions over state-of-the-art methods. Future work includes conformal training for multi-target regression and deployment in healthcare applications.

## 7 Acknowledgments

This work was supported in part by USDA-NIFA funded AgAID Institute, the National Institutes of Health (NIH), and the National Science Foundation (NSF). The views expressed are those of the authors and do not reflect the official policy or position of the USDA-NIFA, NIH, or NSF.

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

# A  Appendix

## A.1  Technical Proof

**Theorem 1.** *Assume the calibration examples $\{(X_i, Y_i)\}_{i \in \mathcal{D}_{cal}}$ and the test example $(X_{test}, Y_{test})$ are exchangeable. Let $V_i$ denote the nonconformity score computed for $(X_i, Y_i)$ (and similarly $V_{test}$ for $(X_{test}, Y_{test})$). For any center $X$, define normalized weights*

$$\bar{w}_j(X) \ := \ \frac{w(X, X_j)}{\sum_{\ell \in \mathcal{D}_{cal}} w(X, X_\ell)}, \qquad j \in \mathcal{D}_{cal},$$

*and the weighted empirical CDF*

$$\hat{F}_X(t) \ := \ \sum_{j \in \mathcal{D}_{cal}} \bar{w}_j(X) \, \mathbf{1}\{V_j \le t\}.$$

*Let the corresponding weighted $\gamma$-quantile be*

$$\hat{q}_X(\gamma) \ := \ \inf\{t : \hat{F}_X(t) \ge \gamma\}.$$

*Choose the corrected global level $\tilde{\alpha}$ according to Eq. 8. Then the APR prediction region $\hat{R}_{\mathcal{Y}}(X_{test})$ constructed using $\hat{q}_{X_{test}}^{\mathrm{APR}} = \hat{q}_{X_{test}}(\tilde{\alpha})$ satisfies the distribution-free finite-sample marginal coverage guarantee*

$$\mathbb{P}\{Y_{\text{test}} \in \hat{R}_{\mathcal{Y}}(X_{\text{test}})\} \ge 1 - \alpha.$$

*Proof.* This result follows directly from the localized/weighted conformal prediction guarantee of Guan (2023) (Theorem 1). Specifically, using the normalized weights $\bar{w}_j(X)$ and the corrected level $\tilde{\alpha}$ in Equation 8, their theorem implies that the conformal set defined by the weighted empirical quantile attains marginal coverage at least $1 - \alpha$ under exchangeability. Instantiating that construction with our nonconformity scores and decoding step yields the stated guarantee for $\hat{R}_{\mathcal{Y}}(X_{\text{test}})$. □

**Theorem 2.** *Suppose all calibration samples $(X_i, Y_i) \in \mathcal{D}_{cal}$ and the test pair $(X, Y)$ are exchangeable. Let $R_{\mathcal{Z}}^k(X)$ be the kNN-localized latent-space prediction region produced by APR (i.e., constructed using weights supported on the $k$ nearest calibration inputs to $X$), and let $R_{\mathcal{Z}}(X)$ be the corresponding non-localized (global) latent-space prediction region obtained without kNN localization. Define the decoded target-space regions*

$$R_{\mathcal{Y}}^k(X) := \mathcal{D}\big(R_{\mathcal{Z}}^k(X)\big) \quad \text{and} \quad R_{\mathcal{Y}}(X) := \mathcal{D}\big(R_{\mathcal{Z}}(X)\big).$$

Assume Assumption 1 and that the decoder $\mathcal{D}$ preserves the relative order of expected volume in the sense of Definition 2. Then APR's kNN-localized target-space regions are no larger in expected volume than the non-localized regions, i.e.,

$$\mathbb{E}_X\big[|R_{\mathcal{Y}}^k(X)|\big] \le \mathbb{E}_X\big[|R_{\mathcal{Y}}(X)|\big].$$

*Proof.* APR constructs latent-space regions using an *isotropic* score based on Euclidean distance in the latent space. Concretely, for each input $X$, the (localized) region has the form of an $r$-dimensional closed ball

$$R_{\mathcal{Z}}^k(X) = \{z \in \mathcal{Z} : \|z - \mu_{\mathcal{Z}}(X)\|_2 \le v^k(X)\}, \qquad R_{\mathcal{Z}}(X) = \{z \in \mathcal{Z} : \|z - \mu_{\mathcal{Z}}(X)\|_2 \le v^{\mathrm{glob}}(X)\},$$

where $\mu_{\mathcal{Z}}(X)$ is the latent center returned by the encoder/decoder pair and $v^k(X)$ (resp. $v^{\mathrm{glob}}(X)$) is the calibrated quantile radius with kNN-localized (resp. global) weights.

The volume of an $r$-dimensional ball with radius $v$ is $c_r v^r$ for a dimension-dependent constant $c_r > 0$. Therefore, by Assumption 1 and the monotonicity of $v \mapsto v^r$ on $\mathbb{R}_+$,

$$|R_{\mathcal{Z}}^k(X)| = c_r \, (v^k(X))^r \le c_r \, (v^{\mathrm{glob}}(X))^r = |R_{\mathcal{Z}}(X)| \quad \text{a.s. in } X.$$

Taking expectations over $X$ gives $\mathbb{E}_X[|R_{\mathcal{Z}}^k(X)|] \leq \mathbb{E}_X[|R_{\mathcal{Z}}(X)|]$. Finally, applying Definition 2 with $\tilde{R}_{\mathcal{Z}}(X) = R_{\mathcal{Z}}(X)$ yields

$$\mathbb{E}_X\big[\,|\mathcal{D}(R_{\mathcal{Z}}^k(X))|\,\big] \leq \mathbb{E}_X\big[\,|\mathcal{D}(R_{\mathcal{Z}}(X))|\,\big],$$

which is exactly

$$\mathbb{E}_X[|R_{\mathcal{Y}}^k(X)|] \leq \mathbb{E}_X[|R_{\mathcal{Y}}(X)|]$$

$\square$

## A.2 Dataset Splits and Hyperparameter `k` for $k$-NN

This section provides further details about the number of data points used for training, testing, validation, calibration (for other methods), and average $k$ calibration inputs (for both variants of APR) across all datasets. To choose $k$ for the test input-conditioned quantile threshold in APR, we performed a systematic search within 30% to 90% of the calibration set (i.e., we sweep $k \in \{0.3m, 0.4m, \ldots, 0.9m\}$ with $m = |\mathcal{D}_{cal}|$) and report the best-performing $k$; we also observe stable behavior across a broad range of $k$ values., selecting the $k$ value that provides the smallest region size. $d$ is the number of target outputs for each multi-target regression task.

| Dataset | Targets | Training | Testing | Validation | Calibration | mean k (%) |
|---|---|---|---|---|---|---|
| BIWI_2 | 2 | 512 | 200 | 160 | 128 | 104.00(82.00%) |
| BIWI_3 | 3 | 512 | 200 | 160 | 128 | 104.00(82.00%) |
| Community_2 | 2 | 1020 | 399 | 319 | 256 | 180.15 (70.4%) |
| Community_3 | 3 | 1020 | 399 | 319 | 256 | 189.15 (73.9%) |
| Community_4 | 4 | 1020 | 399 | 319 | 256 | 203.25 (79.3%) |
| Bio | 2 | 5120 | 2000 | 1600 | 1280 | 1100.80 (86.0%) |
| House | 2 | 11065 | 4323 | 3458 | 2767 | 1259.75 (45.5%) |
| Blog | 2 | 11264 | 4400 | 3520 | 2816 | 1239.10 (44.0%) |
| Maint._2 | 2 | 1024 | 400 | 320 | 256 | 152.80 (60.0%) |
| Maint._3 | 3 | 1024 | 400 | 320 | 256 | 168.15 (65.7%) |
| Maint._4 | 4 | 1024 | 400 | 320 | 256 | 141.20 (55.2%) |

Table 5: Number of training, testing, validation, calibration (for other methods), and the number of APR calibration points $k$ (averaged over 20 runs) across all datasets.

## A.3 Real Experimental Results

**GitHub Repository.** The code necessary for implementing APR and replicating the results of our paper can be found in the following anonymous GitHub repository: `https://anonymous.4open.science/r/apr-4C4C/`

**Compute Machine Specifications:** All experiments were conducted on the following hardware and software setup:

- **Operating System:** Rocky Linux 8.10 (Green Obsidian), **Processor:** AMD EPYC 7573X 32-Core Processor

- **GPUs:** Two NVIDIA A40 GPUs (each with 46 GB of memory), **GPU Driver Version:** 555.42.02, **CUDA Version:** 12.5

| Dataset | Targets | Methods | Coverage | Region Area ↓ | Reduction(%) from Naive ↑ | Reduction(%) from SOTA ↑ |
|---|---|---|---|---|---|---|
| BIWI_2 | 2 | Naive | 0.92 (0.006) | 521.92 (38.24) | — | — |
| | | SOTA | 0.91 (0.013) | 352.69 (56.76) | 32.42% | — |
| | | APR-U | 0.90 (0.013) | 328.67 (51.18) | 37.03% | 6.81% |
| | | APR-W | 0.89 (0.013) | 299.51 (46.15) | **42.61%** | **15.08%** |
| BIWI_3 | 3 | Naive | 0.93 (0.008) | 12373.15 (1429.73) | — | — |
| | | SOTA | 0.92 (0.009) | 5202.63 (951.92) | 57.95% | — |
| | | APR-U | 0.91 (0.010) | 4747.07 (941.92) | 61.63% | 8.76% |
| | | APR-W | 0.90 (0.010) | 4049.22 (771.71) | **67.27%** | **22.17%** |
| Community_2 | 2 | Naive | 0.90 (0.004) | 885.48 (26.13) | — | — |
| | | SOTA | 0.91 (0.005) | 402.07 (12.48) | 54.59% | — |
| | | APR-U | 0.89 (0.006) | 378.31 (10.36) | 57.28% | 5.91% |
| | | APR-W | 0.89 (0.006) | 375.74 (10.26) | **57.57%** | **6.55%** |
| Community_3 | 3 | Naive | 0.90 (0.006) | 21933.56 (1050.76) | — | — |
| | | APR-W | 0.90 (0.007) | 4394.57 (202.64) | 77.94% | — |
| | | SOTA | 0.91 (0.005) | 4838.51 (231.29) | 79.65% | 7.75% |
| | | APR-U | 0.90 (0.006) | 4463.30 (206.48) | **79.96%** | **9.18%** |
| Community_4 | 4 | Naive | 0.90 (0.006) | 35745.29 (2689.90) | — | — |
| | | SOTA | 0.91 (0.004) | 3566.06 (292.63) | 90.02% | — |
| | | APR-U | 0.90 (0.005) | 3160.70 (214.17) | 91.16% | 11.37% |
| | | APR-W | 0.90 (0.005) | 3082.39 (208.17) | **91.38%** | **13.56%** |
| Bio | 2 | Naive | 0.90 (0.002) | 504.75 (7.84) | — | — |
| | | SOTA | 0.90 (0.003) | 432.41 (6.16) | **14.33%** | — |
| | | APR-U | 0.90 (0.003) | 436.11 (6.31) | 13.60% | -0.86% |
| | | APR-W | 0.90 (0.003) | 435.61 (6.31) | 13.70% | -0.74% |
| House | 2 | Naive | 0.90 (0.002) | 384.00 (5.08) | — | — |
| | | SOTA | 0.90 (0.002) | 339.75 (8.73) | 11.52% | — |
| | | APR-U | 0.89 (0.002) | 326.75 (9.89) | 14.91% | 3.83% |
| | | APR-W | 0.89 (0.002) | 326.23 (9.89) | **15.05%** | **3.98%** |
| Blog | 2 | Naive | 0.90 (0.001) | 245.15 (6.65) | — | — |
| | | SOTA | 0.90 (0.002) | 259.81 (11.65) | -5.98% | — |
| | | APR-U | 0.87 (0.003) | 218.82 (12.99) | 10.74% | 15.78% |
| | | APR-W | 0.87 (0.003) | 216.41 (12.91) | **11.73%** | **16.71%** |
| Maint._2 | 2 | Naive | 0.90 (0.006) | 466.12 (173.51) | — | — |
| | | SOTA | 0.99 (0.001) | 147.27 (15.00) | 68.41% | — |
| | | APR-U | 0.86 (0.009) | 21.16 (1.45) | **95.46%** | **85.63%** |
| | | APR-W | 0.95 (0.005) | 21.34 (1.44) | 95.42% | 85.51% |
| Maint._3 | 3 | Naive | 0.91 (0.006) | 1213850.9 (408676) | — | — |
| | | SOTA | 0.98 (0.002) | 3590.52 (752.77) | 99.70% | — |
| | | APR-U | 0.88 (0.008) | 2913.02 (712.51) | 99.76% | 18.87% |
| | | APR-W | 0.94 (0.006) | 2493.11 (641.75) | **99.79%** | **30.56%** |
| Maint._4 | 4 | Naive | 0.91 (0.005) | 53031241 (26324344) | — | — |
| | | SOTA | 0.98 (0.002) | 5957.45 (1298.19) | 99.99% | — |
| | | APR-W | 0.98 (0.006) | 4277.23 (2066.71) | 99.99% | 28.20% |
| | | APR-U | 0.87 (0.010) | 4282.11 (2066.65) | **99.99%** | **28.12%** |

Table 6: Coverage rates, region size, and reduction in region size of APR relative to SOTA method in the space $\mathcal{Y}$ presented for eleven datasets with multiple targets from different areas at $\alpha = 0.1$.

| Dataset | Targets | Methods | Coverage | Region Area ↓ | Reduction(%) from Naive ↑ | Reduction(%) from SOTA ↑ |
|---|---|---|---|---|---|---|
| Bio | 2 | Naive | 0.94 | 715.83 | − | − |
| | | SOTA | 0.95 | 596.02 | 16.74% | − |
| | | APR-U | 0.94 | 587.96 | 17.86% | 1.35% |
| | | APR-W | 0.95 | 587.15 | **17.98%** | **1.49%** |
| BIWI_2 | 2 | Naive | 0.97 | 1266.76 | − | − |
| | | SOTA | 0.96 | 609.81 | 51.86% | − |
| | | APR-U | 0.94 | 526.68 | 58.42% | 13.63% |
| | | APR-W | 0.93 | 487.22 | **61.54%** | **20.10%** |
| BIWI_3 | 3 | Naive | 0.96 | 30841.33 | − | − |
| | | SOTA | 0.96 | 7459.55 | 75.81% | − |
| | | APR-U | 0.95 | 7098.47 | 76.98% | 4.84% |
| | | APR-W | 0.94 | 6036.01 | **80.43%** | **19.08%** |
| Blog | 2 | Naive | 0.95 | 475.96 | − | − |
| | | SOTA | 0.96 | 404.87 | 14.94% | − |
| | | APR-U | 0.93 | 359.69 | 24.43% | 11.16% |
| | | APR-W | 0.93 | 355.85 | **25.24%** | **12.11%** |
| Community_3 | 3 | Naive | 0.96 | 55285.24 | − | − |
| | | SOTA | 0.96 | 9890.66 | 82.11% | − |
| | | APR-U | 0.95 | 8832.62 | 84.02% | 10.70% |
| | | APR-W | 0.95 | 8688.65 | **84.28%** | **12.15%** |
| Community_4 | 4 | Naive | 0.95 | 92377.72 | − | − |
| | | SOTA | 0.96 | 7520.86 | 91.86% | − |
| | | APR-U | 0.95 | 6904.66 | 92.53% | 8.19% |
| | | APR-W | 0.95 | 6735.23 | **92.71%** | **10.45%** |
| Community_2 | 2 | Naive | 0.96 | 1611.00 | − | − |
| | | SOTA | 0.96 | 616.96 | 61.70% | − |
| | | APR-U | 0.95 | 575.77 | 64.26% | 6.68% |
| | | APR-W | 0.95 | 571.18 | **64.55%** | **7.42%** |
| House | 2 | Naive | 0.95 | 554.69 | − | − |
| | | SOTA | 0.95 | 457.18 | 17.58% | − |
| | | APR-U | 0.94 | 428.25 | 22.79% | 6.33% |
| | | APR-W | 0.94 | 427.33 | **22.96%** | **6.53%** |
| Maint._2 | 2 | Naive | 0.94 | 2045.03 | − | − |
| | | SOTA | 0.99 | 169.80 | 91.70% | − |
| | | APR-U | 0.92 | 19.03 | 99.07% | 88.79% |
| | | APR-W | 0.97 | 20.42 | **99.00%** | **87.98%** |
| Maint._3 | 3 | Naive | 0.95 | 4187810.75 | − | − |
| | | SOTA | 0.98 | 5419.40 | 99.87% | − |
| | | APR-U | 0.93 | 1278.50 | 99.97% | 76.41% |
| | | APR-W | 0.97 | 1281.79 | **99.97%** | **76.35%** |
| Maint._4 | 4 | Naive | 0.95 | 238938747.50 | − | − |
| | | SOTA | 0.97 | 12328.73 | 99.99% | − |
| | | APR-U | 0.93 | 1279.99 | 100.00% | 89.62% |
| | | APR-W | 0.99 | 1134.18 | **100.00%** | **90.80%** |

Table 7: Coverage rates, region size, and reduction in region size of APR relative to SOTA method in the space $\mathcal{Y}$ presented for eleven datasets with multiple targets from different areas at $\alpha = 0.05$.

