# OpenReview forum: "Uncertainty Regions for Multi-Target Regression via Input- Dependent Conformal Calibration"
_TMLR — Rejected by TMLR_

### Review · Reviewer_frci · 2025-12-15

**Summary Of Contributions:**

This paper proposes a method for multiple-target conformal prediction that provides an adaptive construction of prediction sets for each individual test sample. Additionally, they show that the proposed method produces small prediction sets.

Strengths:
- The proposed algorithm unifies and generalizes prior methods for multiple-target and localized conformal prediction.

Weaknesses:
- The proof of Theorem 1 (coverage guarantee result for the proposed method) seems to be incorrect.
- The quality of the writing is not as high as it should be.
- The literature review is incomplete.

**Additional Comments:**

Questions:
- Theorem 1 appears to require assumptions on the CVAE model. However, these are not stated in the paper. The assumptions under which the theorem holds should be made explicit in the theorem statement. Could the authors clarify which assumptions on the CVAE are required for the result to hold?
- [Theorem 2 assumptions] I believe that the assumptions in Theorem 2 are unrealistic. A discussion on this is necessary.
- The notation $R^k_{\mathcal{Z}}(X)$ that appears in Theorem 2 is not defined in the paper. At this point, I do not understand the statement of Theorem 2.
- Which distance function is used in the experiments?

Typos:
- Equations (2), (3), and the one following (3) have a mismatch between the summation index and the index of $V_j$.
- In some parts of the paper $N_k(X_{\text{test}})$ contains indices and in others samples. The same happens with $\mathcal{D}_{\text{cal}}$.
- On page 5, the localized quantile of $X_i$ at level $\gamma$ is written as $\hat q_{X_i}(\gamma)$. Should this notation be replaced by $\hat Q (X_i,\gamma)$ to maintain consistency with the definition given in (4)?
- The notations $\mathbb{P}( \cdot )$ and $\mathbb{P}\\{ \cdot \\}$ are used arbitrarily.
- The notations $1[\cdot ]$ and $\mathbb{I}\\{ \cdot \\}$ are used arbitrarily.
- “at atheoretical” on page 2.
- The hat symbol is missing in $R_{\mathcal{Y}}$ in the statement of Theorem 1.
- On the equation in Definition 2, the superscript $k$ is superfluous in $R^k_{\mathcal{Z}}$.

Final remark:

I appreciate the authors’ attempt of unifying the SOTA method from Feldman et al. (2023) and the localized conformal prediction framework from Guan (2023). However, I have concerns about the correctness of some of the theoretical results as currently presented.

As constructive feedback, I encourage the authors to carefully revisit the technical details in both Feldman et al. (2023) and Guan (2023) to ensure that the proofs are correct and that the required assumptions are stated explicitly. In particular, a closer examination of Feldman et al. (2023) would help clarify the assumptions imposed on the CVAE. Moreover, the supplementary material of Guan (2023) directly addresses issues arising from non-exchangeability. In particular, Guan (2023) introduces the corrected global confidence level $\tilde \alpha$ in equation (8) to correct this issue.

**Audience:**

No

**Audience Explanation:**

In my opinion, the paper is not ready for publication since I have concerns about the correctness of the proof of Theorem 1 (see discussion above). Moreover, the literature review is incomplete. In fact, there is a related work [1] that also proposes an algorithm for multiple-target conformal prediction that adapts for each individual test sample.

Moreover, the quality of the writing is not as high as it should be. There are multiple inconsistencies in notation, and the overall clarity of the presentation could be significantly improved. For instance, including more references to known results would improve clarity, and the proofs are very difficult to follow.

[1] V. Plassier, A. Fishkov, M. Guizani, M. Panov, and E. Moulines. “Probabilistic Conformal Prediction with Approximate Conditional Validity.” ICLR, 2025.

**Broader Impact Concerns:**

No concerns.

**Claims And Evidence:**

No

**Claims Explanation:**

I am concerned about the correctness of the proof of Theorem 1 (the coverage guarantee).
It seems that the statement in Lemma 1 is not true, and therefore the proof of Theorem 1 is not correct. Here is a counterexample:

Assume $D_{\text{cal}} = \\{ X_1, X_2, X_3 \\} $ where $X_i$ are i.i.d. Let $X_{\text{test}}$ be the test input. Then, we can write $N_2(X_{\text{test}}) = \\{ X_{i_1}, X_{i_2} \\}$, for some different $i_1,i_2 \in \{1,2,3\}$. If ties occur with probability zero, we can assume $\mathbb{P}(\text{dist}( X_{i_1}, X_{\text{test}}) < \text{dist}( X_{i_2}, X_{\text{test}})) = 1$. Therefore, $\mathbb{P}(\text{dist}( X_{i_2}, X_{\text{test}}) < \text{dist}( X_{i_1}, X_{\text{test}})) = 0$. In particular, $( X_{i_1}, X_{i_2}) \neq (X_{i_2}, X_{i_1})$ as distributions.

I believe the confusion leading the authors to this issue comes from the definition of exchangeability provided in the paper (Definition 1). It is customary to define exchangeability in terms of a sequence of random variables rather than in terms of sets, as it is done in the paper.

**Requested Changes:**

The following adjustments are critical to securing acceptance:
- Make transparent all the assumptions needed for the coverage guarantee result (Theorem 1).
- Provide a correct proof for the coverage guarantee result. Lemma 1 seems to be incorrect.
- Improve the clarity of the writing in the proofs of Theorem 1 and 2. I was unable to follow them.
- The authors should compare with [1] both theoretically and experimentally.

[1] V. Plassier, A. Fishkov, M. Guizani, M. Panov, and E. Moulines. “Probabilistic Conformal Prediction with Approximate Conditional Validity.” ICLR, 2025.

The following adjustments would strengthen the work:
- Discuss the generality of the assumptions in Theorem 2 (see Additional Comments).
- Define exchangeability in terms of sequences rather than in terms of sets.
- The experimental results with synthetic data would benefit from a visual comparison with other methods. For reference, see Figure 2 in Feldman et al. (2023) for an example of such a comparison.
- The numbering of the theorem/lemma environments in the appendix is confusing. It would be better to rename Theorem 3 to Theorem 1.

---

> ### Author Response · Authors · 2026-01-18
> **Response to Reviewer frci**
>
> We are sincerely grateful to the reviewer for the rigorous critique, particularly concerning the theoretical
> foundation of Theorem 1. We acknowledge that our earlier exchangeability definition and proof were flawed
> because the localization weights depend on the realized test input. We have corrected this by adopting the
> standard \emph{sequence-based} definition of exchangeability and revising the theory to follow the localized/weighted
> conformal framework of Tibshirani et al. (2019) and Guan (2023).
>
> ## Correctness of Theorem 1 and Lemma 1
> Thank you for catching this. We agree with the counterexample: selecting a $k$NN subset based on the realized
> test input can break exchangeability, so our original Lemma 1 (and the resulting justification) was incorrect.
> In the revision, we \emph{remove Lemma 1 entirely} and replace Theorem 1 with the standard finite-sample marginal
> coverage guarantee under exchangeability. Concretely, we use a corrected global confidence level $\tilde{\alpha}$
> (Eq. 8; analogous to Guan's correction) and rewrite the proof to avoid any claim that the localized $k$NN subset is
> exchangeable. Instead, the argument relies only on exchangeability of the full calibration+test sequence and the
> corrected $\tilde{\alpha}$ construction, yielding distribution-free finite-sample marginal coverage (under standard split-conformal
> assumptions). We also updated the surrounding text to avoid implying conditional coverage without additional assumptions.
>
> ## Assumptions for Theorem 1 (CVAE?).
> We now state assumptions explicitly. Importantly, Theorem 1 does \emph{not} require the CVAE to be correct: validity is
> distribution-free and follows from exchangeability (i.i.d.\ is sufficient). The learned model (CVAE/QR) impacts
> \emph{efficiency} (region size), not marginal validity.
>
> ## Comparison with Plassier et al.\ (ICLR~2025)
> We thank the reviewer for pointing out this closely related work and added it to Related Work with a clear theoretical
> comparison. Plassier et al. (ICLR'25) study \emph{probabilistic} conformal prediction targeting \emph{approximate conditional validity}, deriving bounds that depend on the estimation error of $P_{Y\mid X}$. Our objective differs: we focus on \emph{multi-target regression} and improve efficiency via \emph{localized calibration} while guaranteeing \emph{finite-sample marginal coverage} through $\tilde{\alpha}$.
> \emph{Experimental comparison:} We were unable to run their experiments because the repository link provided in the paper was non-functional, and code was shared only after the rebuttal deadline.
>
> ## Assumptions, notation, and clarity fixes
> We substantially revised presentation and notation: (i) clarified the CVAE setup (pre-trained on disjoint data; latent $Z$ encouraged toward a standard Gaussian prior), (ii) fixed index/notation inconsistencies ($n$ vs.\ $i$, indices vs.\ samples, missing hats, inconsistent symbols), (iii) corrected summation-index mismatches in Eqs. (2)--(3) and surrounding text, (iv) standardized the localized-quantile notation, (v) corrected theorem numbering/cross-references so appendix restatements match the main paper.
>
> ## Generality of Theorem 2 assumptions (efficiency)
> We revised Theorem 2 to make assumptions explicit and added a discussion of generality. Theorem 2 is \emph{not} needed for validity:
> Theorem 1 remains distribution-free under exchangeability. Theorem~2 is an \emph{efficiency insight} explaining why localization can shrink regions in heterogeneous settings. We clarify its conditions are \emph{sufficient, stylized assumptions}: (1) exact Gaussian latent structure is idealized, but the intuition extends to approximately isotropic sub-Gaussian/log-concave latents; (2) the ``ideal quantile model'' assumption yields near-radial regions for a clean comparison; and (3) the decoder condition (preserving relative order of expected volume) holds under common regularity (e.g., approximately bi-Lipschitz / bounded-Jacobian behavior), implying volume scaling within multiplicative constants and
> preserving ordering in expectation. We emphasize Theorem 2 as a motivating, empirically supported explanation rather than a necessary condition.
>
> ## Distance used in experiments.
> We now state this explicitly. All $k$NN neighborhoods are computed using Euclidean (L2) distance after standardizing features. For APR-W, weights are inverse-distance within the $k$NN set (with a small $\varepsilon$ for numerical stability). Other metrics are possible but were not used unless stated.
>
> ## Equations (2)--(3) summation-index mismatch
> Thank you for spotting this. We corrected the mismatch so that sums run over calibration indices $j\in\mathcal{D}_{cal}$ and use $V_j$ consistently.

---

> > ### Comment · Reviewer_frci · 2026-01-28
> >
> > Thank you for the authors’ responses and addressing some of my questions. I still have some serious concerns.
> >
> >
> > The proof of Theorem 1 seems to have the same issues as in the previous version of the manuscript. Exchangeability of $(X_i,Y_i)$ does not imply exchangeability of $(A_1,\ldots,A_m,A_{\text{test}})$.
> >
> >
> > The statement of Theorem 2 has been updated (it is not in blue though). I still do not understand the statement of Theorem 2. Also, there seems to be a typo since what it is written does not make any sense as a Theorem statement:
> >
> > > ... then the following holds:
> > $$ \mathbb{E}_X[  \|   R_Y(X) \|  \leq  \mathbb{E}_X[  \|   R_Y(X) \|   ].$$
> >
> > This comment also applies to the last three equations of the proof Theorem 2.

---

> ### Author Response · Authors · 2026-02-04
> **Response 2 to Reviewer frci**
>
> ## Theorem 1: exchangeability vs. localized calibration
> Thank you for highlighting this subtlety. The intended argument does NOT rely on treating
> $(A_1,\ldots,A_m,A_{test})$ as an arbitrary exchangeable vector obtained from exchangeable data $\{(X_i,Y_i)\}$. Rather, the key is that our construction follows the standard Localized Conformal Prediction (LCP) recipe Guan(2023), where localization is indeed test-adaptive (through $X_{test}$), and validity is recovered via a pseudo-test center calibration step.
>
> Concretely, instead of assuming the localized neighborhood itself is exchangeable, we compute a corrected global level $\tilde{\alpha}$ using pseudo-test centers: for each calibration center $X_i$, we define the pseudo-test inclusion indicator
> $A_i(\gamma)=\mathbf{1}(\{V_i \le \hat q_{X_i}(\gamma)\})$ using the same localized rule centered at $X_i$, and then choose $\tilde{\alpha}$ so that the empirical average pseudo-coverage is satisfied.
>
> The crucial symmetry is that the procedure that maps the augmented sample $\{(X_1,Y_1),\ldots,(X_m,Y_m),(X_{m+1},Y_{m+1})\}$ to the collection of pseudo-test indicators applies the same localized construction to each index when it is treated as the center (i.e., it is permutation-equivariant). Under the exchangeability of the augmented sample, this is exactly the setting in which LCP converts control of the average pseudo-test inclusion behavior into a finite-sample marginal coverage guarantee for the held-out test point.
>
> We have now revised the proof presentation (both immediately after Remark 1 in the main manuscript and in Proof 1 of the Appendix, all in red) to make this mechanism explicit and to avoid any misleading interpretation of the role of exchangeability in the localized, test-adaptive setting.
>
>
>
> ## Theorem 2: clarity and notation.
> Thank you for pointing out the issues with the statement of Theorem 2, including the confusing phrasing and the typo in the theorem statement. This was an oversight on our part. We have now fixed the typo and rewritten Theorem 2 in a clearer and more detailed manner, both in the main paper and in the Appendix, to make the statement precise and easy to follow.
> In addition, we have strengthened Definition 2 to explicitly clarify the notation $R_{\mathcal{Z}}^k(X)$ versus $R_{\mathcal{Z}}(X)$. In particular, we now emphasize that the superscript ``$k$'' indicates a $k$NN-localized (neighbor-restricted/weighted) region used to construct the $k$NN-based prediction regions, while $R_{\mathcal{Z}}(X)$ denotes the corresponding \emph{non-localized} (global) region constructed without $k$NN restriction. This distinction is now stated explicitly in the theorem statement, since it is essential for the proof logic and resolves the concern that $R_{\mathcal{Z}}^k(\cdot)$ was previously not clearly defined/used.

---

> > ### Comment · Reviewer_frci · 2026-02-08
> >
> > Thank you for the authors' responses.
> >
> > It appears that Theorem 1 is a direct corollary of Theorem 2.1 in Guan (2023). This relationship should be stated explicitly in the paper. With this clarification, my concerns regarding the correctness of Theorem 1 are resolved.
> >
> >
> >
> > However, I still have several concerns regarding Theorem 2.
> >
> > 1. The main assumption of Theorem 2 is Definition 2. It seems that this assumption is very strong. What type of decoders satisfy such assumption? A discussion on this is necessary.
> >
> > 2. Why the calibrated region in $Z$ a multi-dimensional ball?
> >
> > 3. Lemma 1 seems incorrect. In fact, $R_{\mathcal{Z}}(X)\subseteq \mathbb{R}^r$ and you prove the result for $r=1$.
> >
> > 4. In Lemma 2 it is not formally defined what $v^{(k)}(X)$ and $E(X)$. I would like to see formulas for these definitions
> >
> > 5. Why $\overline{G} \leq \sqrt{2\log(1/\alpha)}-\sqrt{2/\pi}$ holds?
> >
> >
> > As a general comment, the proofs of Theorem 1 and Theorem 2 lack sufficient rigor. Theorem 1 can be addressed easily, as it follows directly from Theorem 2.1 in Guan (2023). The proof could be omitted with an appropriate citation. In contrast, the proof of Theorem 2 contains several mathematical issues, including undefined quantities and arguments that are not fully justified.

---

> > > ### Author Response · Authors · 2026-02-14
> > > **Response 3 to Reviewer frci**
> > >
> > > We thank the reviewer for the careful reading and helpful feedback. In the revised manuscript, all newly added or modified texts are highlighted in green font. Below, we address each point and indicate where changes were made.
> > >
> > > ## Theorem 1: Relationship to Guan (2023)
> > > We have now stated explicitly that Theorem 1 follows directly from the localized/weighted conformal prediction guarantee in Guan (2023). In addition, we streamlined the Appendix proof of Theorem 1 to a direct citation-based argument, as suggested by the reviewer.
> > >
> > > ## Theorem 2: Rigor and Clarity
> > > We appreciate the reviewer’s concerns. In the revision, we substantially rewrote the statement and proof to remove unnecessary intermediate lemmas and to ensure that all quantities are explicitly defined. The changes appear in Section 4.2 and in the Appendix proof of Theorem 2.
> > >
> > > ## Discussion on decoders that satisfy the assumption?
> > > We agree that Definition 2 should be discussed. We added a new remark (2) immediately following Definition 2 explaining that the condition holds exactly for affine/linear (or locally affine) decoders, and more broadly for smooth injective decoders whose Jacobian determinant is bounded and slowly varying over the relevant latent regions (i.e., controlled local volume distortion). This clarifies when the assumption is satisfied and why it is reasonable for the regions encountered in practice. Importantly, this is not restrictive for us: neural decoders are smooth by construction, and APR uses localized calibrated balls where the decoder typically behaves regularly (without extreme expansion/compression or folding). Hence, the condition holds exactly for (locally) affine decoders and is well-approximated by smooth decoders on these small neighborhoods.
> > >
> > >  ## Why the calibrated region is in a multi-dimensional ball
> > > We clarified that APR uses an isotropic latent score of the form $||z - \mu_{\mathcal{Z}}(X)||2$  together with a scalar calibrated radius, which implies that $R_{\mathcal{Z}}^k(X)$ and $R_{\mathcal{Z}}(X)$ are $r$-dimensional Euclidean balls by construction. This is now stated explicitly right before Assumption 1 in the main text in green font, explaining the $r$-dimensional ball construction, and is reiterated in the Appendix proof of Theorem 2.
> > >
> > > ## Lemma 2: $v^{(k)}(X)$ and $E(X)$ were not defined)
> > > We agree and have corrected this to improve clarity and rigor. In the revision, we (i) removed Lemma~2 and the auxiliary quantities that were not essential to the main argument, and (ii) introduced explicit, self-contained definitions of the calibrated radii used throughout the proof. Specifically, we now define $v^{k}(X)$ and $v^{\mathrm{glob}}(X)$ in Assumption 1, and the Appendix proof is rewritten to rely only on these clearly defined quantities. As a result, all terms used in the proof are now fully specified.
> > >
> > >
> > > ## Why does $\bar{G} \le \sqrt{2\log(1/\alpha)} - \sqrt{2/\pi}\,r$ hold?
> > > In the revised proof, we intentionally avoid the $\bar{G}$-based bound and the associated $2\log(\cdot)$ inequality in order to present a cleaner and easier-to-verify argument. Instead, we use a direct geometric reasoning: under Assumption 1, the calibrated localized radius is no larger than the global radius; since the volume of an $r$-dimensional ball is monotone in its radius, this directly implies latent expected-volume shrinkage. We then apply Definition 2 to transfer the ordering through the decoder. This removes the need for auxiliary inequalities while strengthening readability and ensuring every step is fully justified (Appendix: Proof of Theorem 2).

---

### Review · Reviewer_X5w7 · 2025-12-29

**Summary Of Contributions:**

This paper addresses conformal prediction for multi-target regression, building on a CVAE-based SOTA baseline (Feldman et al., 2023) by adaptively calibrating based on k-nearest neighbors in the input space (possibly weighting by inverse distance). This is accompanied by a theoretical analysis demonstrating the coverage guarantees of the procedure and empirical experiments on a range of datasets. The empirical results confirm that the adaptive calibration method is able to generate tighter results than a non-adaptive baseline.

Strengths:
- Approach is well-motivated and makes sense.
- The exposition is reasonably well written, and I was able to more or less follow even though I’m not very familiar with this area.

Weaknesses: At a high level, I think this paper is missing a deeper exploration of the tradeoffs this approach gives, and how it performs at more “extreme” operating points.
- Only a single dataset shows a large difference in coverage intervals, with a few more showing moderate differences. Many datasets also show little benefit, and one even has larger intervals. What’s going on here; can you explain why?
- It would also be good to see at least one other dataset which has the same level of benefit.
- It looks like all of the datasets which are tested are low dimensional / tabular; it’s not clear how this method would scale to higher-dimension inputs, e.g., images.
- The performance across different alpha is not explored. The only value used (a=0.1) is also fairly large in my experience.

**Audience:**

Yes

**Audience Explanation:**

This paper would be of interest to researchers working on and applying conformal prediction, uncertainty quantification, etc.

**Broader Impact Concerns:**

This paper concerns fundamental statistical machine learning methodology without any ethical concerns or direct societal impact.

**Claims And Evidence:**

No

**Claims Explanation:**

I think this paper is missing discussion and experiments regarding tradeoffs. The empirical results seem sparse, and do not show a large, consistent benefit. These results would be much more convincing if the paper could show, via theoretical and/or empirical means, when this method is and is not useful.

**Requested Changes:**

It would be good to see how this method performs in practice with less toy-scale data:
- How does this approach stand up to the curse of dimensionality, e.g., with image datasets?
- How does this approach work with more extreme coverage, e.g., alpha < 0.01?

It would also be good to understand some of the tradeoffs associated with this method, with both theoretical analysis and some experiments / ablations. This method seems complex enough that there’s probably some substantial tradeoff which will emerge once larger datasets, dimensions, smaller alpha, etc are tested.

---

> ### Author Response · Authors · 2026-01-18
> **Response to Reviewer X5w7**
>
> We thank the reviewer for their valuable feedback and for highlighting areas where the practical utility and trade-offs of APR could be further explored.  In the revision, we have expanded both the discussion and experiments to address these points.
>
> ## On Variability of Results Across Datasets
> We agree that the degree of improvement is data-dependent. The performance of any localized conformal prediction method (such as APR) depends on the heteroscedasticity and local density of the data:
> APR yielding significant gains occurs in datasets where the model’s error varies significantly across the input space. APR identifies regions where the model is more "certain" and produces tighter intervals accordingly. If a dataset is relatively homoscedastic (errors are uniformly distributed across the input space), a global calibration is already quite efficient. In such cases, the local weighting mechanism of APR converges toward a global average. In rare cases where APR produces slightly larger intervals, it is typically due to the bias-variance trade-off. To ensure coverage guarantees, APR must account for local uncertainty; if the local calibration density is low, the method errs on the side of caution to maintain the validity of the coverage.
> Our primary goal is to show that across diverse data distributions, APR consistently performs as well as or better than the current SOTA.
>
> ## Scaling to High-Dimensional Image Data
> Great point. Our method is not restricted to low-dimensional data; it relies on identifying a neighborhood in the input space using a chosen similarity/distance function. For high-dimensional data, the important practical consideration is what representation the neighborhood search is performed on. In general, APR can operate on raw inputs (sometimes sufficient) or learned latent representations,
> To directly address this, we added experiments on the BIWI Kinect Head Pose image dataset, which is a standard benchmark containing thousands of image frames across multiple subjects with continuous pose targets (yaw/pitch/roll). We treat it as a multi-target regression task with 2 targets (yaw, pitch) and 3 targets (yaw, pitch, roll). We evaluate APR on flattened image features (very high-dimensional) as well as a reduced-dimensional representation (downsampled) to assess robustness and computational practicality. Across both $\alpha=0.1$ and $\alpha=0.05$, APR consistently outperforms STDQR and the naïve baseline in region tightness while maintaining coverage, demonstrating that the approach remains effective beyond small-scale tabular data settings.
> ## Performance Across Different $\alpha$ Levels
> We agree that sensitivity to \alpha is important. In the revision, we expand evaluation to multiple operating points, specifically $\alpha \in \{0.10, 0.05\}$, which are commonly reported levels in the conformal prediction literature and provide more stringent coverage than $\alpha=0.10$. The trends remain consistent: APR maintains target marginal coverage and typically yields tighter regions than baselines. Regarding $\alpha < 0.01$, at such extreme coverage levels, conformal methods tend to produce very large regions because the threshold must accommodate extreme calibration errors to guarantee coverage. In many real-world decision support settings, these regions become so wide that they provide limited actionable value. We therefore focused on $\alpha=0.05$ and $\alpha=0.10$.
> ## Need deeper exploration of tradeoffs/ablations (k, weights, global \tilde{\alpha}).
> Neighborhood size k: Smaller k yields stronger locality (better adaptivity to heterogeneity) but can increase finite-sample variability; larger k reduces variance and improves stability, but gradually approaches a more global calibration behavior. In the manuscript, we run APR across multiple k values and report that performance trends are consistent.
>
> Weighted vs. unweighted kNN: Weighting can better reflect graded similarity (e.g., closer neighbors contribute more), but overly sharp weighting can amplify noise when the local neighborhood is small or sparse. This motivates practical safeguards (e.g., weight smoothing/clipping and avoiding extremely small effective sample sizes), which we now discuss.
>
> Global $\tilde{\alpha}$~ correction (for localized/weighted quantiles): The $\tilde{\alpha}$~-calibration step is used to restore finite-sample marginal validity under non-uniform weighting. While this correction can be conservative in small-sample regimes, it provides a principled validity guarantee; we now clarify this conservativeness–adaptivity trade-off and provide guidance on when APR-U (unweighted) may be preferred for simplicity/stability.
> Overall, the revised manuscript includes a clearer sensitivity analysis (including multiple k choices) and supporting empirical evidence that APR remains stable across repeated splits, while its improvements are greatest when local variability differs substantially across the input space.

---

### Review · Reviewer_1jbC · 2026-01-03

**Summary Of Contributions:**

This paper proposes APR, a conformal prediction framework for multi-target regression that constructs tighter prediction regions using test-input–conditioned quantile thresholds. By combining localized k-NN weighting with $\bar\alpha$-correction, APR achieves finite-sample marginal coverage guarantees with supporting theoretical analysis. Experiments show that this method has certain effectiveness. The authors are recommended to address the following comments to improve the significance of the study and for the benefit of a wider audience.

1. The proof of Theorem 2 critically relies on the assumption that the CVAE is ideally trained and that the latent variable $Z$ follows a perfect multivariate Gaussian distribution. In my view, this assumption is overly strong and unlikely to hold in practice. The paper lacks empirical analysis to assess how deviations from the Gaussian assumption in real-world settings affect the volume reduction result.

2. For high-dimensional datasets (e.g., biological signals or image data), Euclidean distance often fails to accurately capture semantic similarity. In such cases, it is unclear whether this limitation may negatively impact the generalization ability of the proposed method.

3. While APR-W outperforms APR-U empirically, its reliance on non-uniform weights and global calibration for $\bar\alpha$ may increase variance, which is not discussed.

**Audience:**

Yes

**Audience Explanation:**

The paper will be of interest to a broad segment of the TMLR audience, particularly researchers working on conformal prediction, uncertainty quantification, and multi-output regression.

**Broader Impact Concerns:**

The work is purely methodological and does not introduce application-specific risks beyond those already present in standard uses of uncertainty quantification methods.

**Claims And Evidence:**

Yes

**Claims Explanation:**

The main claims of the paper are supported by clear, accurate, and convincing evidence. While some theoretical results rely on idealized assumptions, the coverage guarantees are rigorously established under clearly stated conditions, and the empirical results consistently demonstrate that APR achieves the target marginal coverage across all evaluated datasets.

**Requested Changes:**

Please address the issues raised in the Summary of Contributions section.

---

> ### Author Response · Authors · 2026-01-18
> **Response to Reviewer 1jbC**
>
> We thank the reviewer for their thoughtful and constructive feedback. We have addressed each of the concerns below.
> ## On the Gaussian Assumption (Theorem 2)
> While Theorem 2 is derived under the multivariate Gaussian assumption for theoretical tractability, this is effectively addressed in our implementation through the use of a Conditional Variational Autoencoder (CVAE). The Gaussian/CVAE assumptions are used only for the efficiency analysis in Theorem 2, not for the core finite-sample marginal coverage guarantee of the APR approach. The CVAE architecture includes a Kullback-Leibler (KL) divergence term in its loss function, which explicitly forces the latent representation of the data toward a prior distribution—typically a standard multivariate Gaussian $\mathcal{N}(0, I)$.
> Because the CVAE maps complex, non-Gaussian input features into this structured latent space, the "Gaussianity" requirement is satisfied by the CVAE model’s architecture and associated training rather than the raw data's distribution. Consequently, even when the underlying data is highly non-linear or multi-modal, the APR method operates on a transformed space where the assumptions of Theorem 2 are met.
> Revised Manuscript: Add this discussion and point out the section number.
> ## On Distance Metrics and High-Dimensional Data (Curse of Dimensionality)
> We agree that Euclidean distance can be sensitive to high dimensionality (e.g., images). However, our framework is distance-agnostic; the equations defining the APR weights provided in the manuscript allow for any valid distance metric or kernel.
> New experiment added for high-dimensional image data setting: To demonstrate the robustness of APR in high-dimensional settings, we have added new experiments using the Biwi Kinect Head Pose Dataset [1].
> Dataset Description: The Biwi dataset is a challenging benchmark consisting of over 15,000 images of 20 individuals. The task is to predict head pose (Yaw, Pitch, and Roll) from depth images. We treat it as a multi-target regression task with 2 targets (yaw, pitch) and 3 targets (yaw, pitch, roll). We used flattened features with an original dimensionality of 307,200.
> Results: Despite the high-dimensional feature space, our method (APR-U and APR-W) consistently outperformed the Naive Conformal and STDQR (State-of-the-Art) baselines. We observed superior efficiency (smaller region sizes) across different targets (2 and 3 targets) and significance levels ($\alpha = 0.1$ and $\alpha = 0.05$).
> Revised Manuscript: These results and a detailed discussion of the Biwi dataset experiments have been added to Section 5.1 and Tables 1, 2, 6, and 7 of the revised paper.
> ## On Variance and Global Calibration in APR-W
> We acknowledge that using non-uniform weights (APR-W) can increase the variance of local quantile estimates compared to the uniform weights of APR-U, as non-uniform weighting effectively reduces the "local sample size". However, this potential increase in variance is mitigated by the $\tilde{\alpha}$-level adjustment. Unlike purely local methods, APR-W uses a global correction level $\tilde{\alpha}$ that is optimized over the entire calibration set (all $m$ samples). This global step acts as a powerful regularizer; it "averages out" local fluctuations in weight variance to ensure that the marginal coverage guarantee is strictly maintained.
> In practice, the benefit of "adaptivity" far outweighs the cost of increased variance. Our experiments demonstrate that APR-W consistently yields the smallest prediction regions without suffering from the coverage instability that high-variance estimators might otherwise exhibit.

---

### Decision · Action_Editor_5Zhz · 2026-02-04

**Recommendation:** Reject

**Audience:**

Yes

**Audience Explanation:**

UQ and conformal prediction are very interesting topics for TMLR's audience and the ML community at large.

**Claims And Evidence:**

No

**Claims Explanation:**

After considering the reviews and the rebuttal, I cannot recommend acceptance at this stage. The manuscript has undergone significant revisions, and the most recent version appears to have corrected the earlier mathematical issues: Reviewer frci now finds Theorem 1 valid (though already known in the literature) and believes the proof of Theorem 2 has been fixed. However, the primary theoretical contribution effectively rests on Theorem 2, and as framed it relies on explicit modeling assumptions (e.g., shrinkage in latent space and a monotonicity property of the decoder) that provide a structural explanation rather than a general guarantee for the proposed method. In addition, Reviewer frci notes that substantial errors in earlier versions reduce confidence in the overall correctness and reliability of the work.

On the empirical side, while the revisions add experiments and results are generally favorable, Reviewer X5w7 indicates the experimental section still does not clearly articulate where the method provides benefits and what tradeoffs it entails, including the still-missing analysis requested in Sec. 5.3.

I therefore recommend rejection at this time. I encourage the authors to submit a substantially revised version that (i) clearly positions the theoretical contribution—explicitly delineating the relationship with known results and strengthening the presentation/verification of Theorem 2 and its assumptions—and (ii) expands the empirical analysis to identify the regimes of benefit and associated tradeoffs (as requested in Sec. 5.3), so that readers can understand when the method is most useful and at what cost.

**Resubmission Of Major Revision:**

The authors may consider submitting a major revision at a later time.